# Impact of wildfire smoke on Arctic cirrus formation, part 1: analysis of MOSAiC 2019-2020 observations

Albert Ansmann[1], Cristofer Jimenez[1], Johanna Roschke[1], Johannes Bühl[1,2], Kevin Ohneiser[1], Ronny Engelmann[1], Martin Radenz[1], Hannes Griesche[1], Julian Hofer[1], Dietrich Althausen[1], Daniel A. Knopf[3], Sandro Dahlke[4], Tom Gaudek[1], Patric Seifert[1], and Ulla Wandinger[1]

[1]Leibniz Institute for Tropospheric Research, Leipzig, Germany
[2]Harz University of Applied Sciences, Wernigerode, Germany
[3]School of Marine and Atmospheric Sciences, Stony Brook University, Stony Brook, NY 11794, USA
[4]Alfred Wegener Institute, Helmholtz Centre for Polar and Marine Research, Potsdam, Germany

**Correspondence:** A. Ansmann
(albert@tropos.de)

**Abstract.**

The potential impact of wildfire smoke on Arctic cirrus formation is discussed based on lidar and radar observations during the winter half year of the one-year MOSAiC (Multidisciplinary drifting Observatory for the Study of Arctic Climate) expedition. Aerosol and ice cloud observations were performed aboard the icebreaker *Polarstern* at latitudes >85°N. Aged Siberian wildfire smoke polluted the tropopause region over the central Arctic during the entire winter half year 2019-2020. The smoke particle surface area concentration at the tropopause was of the order of 5-15 $\mu$m$^2$ cm$^{-3}$ and indicated considerably enhanced levels of aerosol pollution for more than six months. Numerous cirrus systems with cloud top temperatures between $-60$ and $-75°$C developed in the polluted upper troposphere. We analyzed all MOSAiC winter cirrus layers regarding their geometrical and optical properties and a subgroup of 20 cirrus events regarding ice water content (IWC) and ice crystal number concentration (ICNC). In the ice virga, which are connected to individual ice nucleation events, ICNC typically ranged from 1-10 L$^{-1}$, was frequently also as high as 20-50 L$^{-1}$, however, observations >100 L$^{-1}$ were rare. Three observational facts corroborate our hypothesis, that smoke significantly influenced Arctic cirrus formation: (a) the occurrence of a long-lasting, persistent smoke pollution layer in the upper troposphere so that favorable conditions for heterogeneous ice nucleation on smoke particles were always given and, at the same time, homogeneous freezing of background aerosol was probably widely suppressed, (b) the high smoke particle surface area concentrations, high enough to significantly trigger ice nucleation on smoke particles as shown in part 2, and (c) the frequently found maximum cirrus ice saturation ratios of 1.3-1.5, which point to the dominance of heterogeneous ice nucleation processes, initiated by inefficient ice-nucleating particles (INPs), as expected when aged smoke particles (i.e., organic aerosol particles) serve as INPs. The studies are continued in the simulation part 2.

## 1 Introduction

A significant increase in the occurrence frequency of wildfire smoke layers in the upper troposphere and lower stratosphere (UTLS) has been observed in the northern hemisphere since 2017 (Baars et al., 2019; Ohneiser et al., 2021; Kloss et al., 2019;

Trickl et al., 2024). The increase may be linked to climate change (Jolly et al., 2015; Abatzoglou et al., 2019; Kirchmeier-Young et al., 2019; Cunningham et al., 2024). In order to adequately consider smoke particles in atmospheric modeling, the role of wildfire smoke in the climate system and respective impact pathways need to be explored in detail. A relevant, climate-sensitive

pathway is ice formation in the upper troposphere. Two ice nucleation modes have to be distinguished, heterogeneous ice nucleation on solid surfaces of ice-nucleating particles (INPs) such as glassy smoke particles and homogeneous ice nucleation of liquid background aerosol particles (sulfate particles). Heterogeneous ice nucleation starts at lower ice saturation ratios than homogeneous freezing. The different nucleation modes may lead to different cirrus properties (crystal size and number concentration) and thus may influence the radiation field and seeder-feeder and precipitation features in the troposphere in

different ways (DeMott et al., 2010). Therefore, aerosol conditions, including wildfire smoke and long-range transport of the pollution, must be well considered in cloud formation parameterizations in regional and global models to allow more accurate weather and climate predictions (Lohmann and Neubauer, 2018; Beer et al., 2022, 2024).

Clear evidence for the ability of aged wildfire smoke particles (organic particles) to initiate heterogeneous ice nucleation was recently provided by Mamouri et al. (2023). Heterogeneous ice nucleation in Californian wildfire smoke was observed with li-

dar in the upper troposphere over the Eastern Mediterranean in October 2020. During the one-year MOSAiC (Multidisciplinary drifting Observatory for the Study of Arctic Climate) expedition (Shupe et al., 2022), we also observed a large number of cirrus systems developing in the wildfire-smoke-polluted upper troposphere over the central Arctic. A first MOSAiC case study of smoke-cirrus interaction was presented by Engelmann et al. (2021). The wildfire smoke originated from record-breaking forest fires in central and eastern Siberia in the summer of 2019 (Ohneiser et al., 2021; Ansmann et al., 2024). The aerosol pollution

spread all over the Arctic in August 2019 (Xian et al., 2022a, b) and reached even the lower stratosphere where the aerosol particles probably contributed to polar ozone depletion in the spring of 2020 (Ohneiser et al., 2021; Voosen, 2021; Ansmann et al., 2022). The UTLS wildfire smoke layer was observable until May 2020 at high northern latitudes.

In this part 1 of a series of two articles, we present an extended analysis of the entire MOSAiC cirrus data set. Goal of our MOSAiC cirrus studies is to provide observational evidence that aged wildfire smoke (organic aerosol particles) significantly

influenced cirrus formation in the central Arctic during the winter half year of 2019-2020. Aim of the simulations, presented in part 2, is to gain a detailed insight into the ability of smoke particles to cause ice nucleation at the observed meteorological and environmental conditions (temperature, relative humidity, updraft speed and amplitude, smoke INP number concentration). The vertical movements, required to initiate ice nucleation, may be caused, e.g., by gravity wave activity (Haag and Kärcher, 2004; **?**; Podglajen et al., 2016; Kärcher and Podglajen, 2019; Kärcher et al., 2019). For the first time observations of aerosol

and cirrus properties are closely combined with comprehensive modeling of cirrus evolution processes to explore smoke-cirrus interaction. Similar studies based on the combination of observations and simulations were conducted after the Eyjafjallajökull eruption in 2010 to investigate the impact of fresh volcanic ash on cirrus formation (Seifert et al., 2011; Rolf et al., 2012).

The MOSAiC smoke and cirrus observations were performed with lidar and radar instruments (Engelmann et al., 2021) aboard the German research ice breaker *Polarstern* (Knust, 2017). The *Polarstern* was trapped in the pack ice and drifted

through the Arctic Ocean from 4 October 2019 to 16 May 2020, mostly at latitudes >85°N. The remote sensing instruments were continuously operated (around the clock) side by side to collect tropospheric and stratospheric aerosol and cloud profile

data up to 30 km height throughout the entire expedition period. Accompanying radiosondes, launched every 6 hours, provided dense sets of observations of the atmospheric state in terms of temperature, relative-humdity, and wind profiles (Maturilli et al., 2021, 2022).

Smoke particles (or more general, organic aerosol particles) seem to be not very efficient INPs (Knopf et al., 2018; Knopf and Alpert, 2023). However, they can influence ice nucleation in very different ways. If the particles are in a glassy state, they can act as INPs in deposition ice nucleation (DIN) processes (Murray et al., 2010; Wang and Knopf, 2011; Wang et al., 2012). DIN is defined as ice formation occurring on the INP surface by water vapor deposition from the supersaturated gas phase. When the smoke particles can take up water and a liquid surface around the particles develops, immersion freezing

can proceed (Wang et al., 2012; Knopf and Alpert, 2013; Knopf et al., 2018). According to classical nucleation theory and observations, DIN INPs are expected to be more efficient in warm cirrus with cloud top temperatures around $-50°$C than in cold cirrus with top temperatures of $-70°$C (Trainer et al., 2009; Pruppacher and Klett, 2010; Alpert et al., 2011; Wang and Knopf, 2011; Wang et al., 2012; Primm et al., 2017). The ice nucleation onset ice saturation ratio $S_{i,on}$ decreases with increasing temperature. Strong, burst-like ice nucleation sets in when the ice saturation ratio $S_i > S_{i,on}$ and is terminated by

ice crystal growth which leads to $S_i < S_{i,on}$ again within a short time period. Theses ice nucleation processes are discussed in the simulation study (part 2). Because of the complex chemical, microphysical, and morphological properties of aged fire smoke particles the development of smoke INP parameterization schemes is generally a crucial task (Knopf et al., 2018).

The particles and released vapors in biomass burning plumes undergo chemical and physical aging processes on their way up to the tropopause and during long-range transport in the UTLS over weeks and months. These aging processes change

the chemical composition of the smoke particles, their morphological characteristics (size, shape, and internal structure), and the internal mixing state of the smoke particles. After finalizing the aging process, the smoke particle may show a core-shell structure with a black-carbon-containing core and an organic-carbon-rich shell, and that the ability to serve as INP mainly depends on the material in the shell and thus on the organic material of the particles. Biomass-burning particles also contain humic-like substances which represent large macromolecules that could serve as INP at low temperatures of $-50$ to $-70°$C

(Kanji et al., 2008; Wang and Knopf, 2011; Wang et al., 2012; Knopf et al., 2018). Jahn et al. (2020) and Jahl et al. (2021) hypothesized that aged smoke particles contain minerals and that these components may determine the smoke INP efficacy.

The article is organized as follows. In Sect. 2, the field campaign, used instrumentation, and the applied lidar and radar data analysis methods are outlined. The key findings of our MOSAiC cirrus studies, which include a cirrus statistical analysis and discussions of case studies, based on the observed smoke and cirrus optical and microphysical properties, are presented in

Sect. 3. Sect. 4 provides a summary and concluding remarks.

## 2 *Polarstern* route, remote sensing, radiosonde and disdrometer instrumentation, and observational products

The MOSAiC expedition began end of September 2019 and lasted until the beginning of October 2020. In this article, we focus on the cirrus observations during the winter half year from the beginning of October 2019 to end of March 2020. Figure 1 shows the track of the drifting *Polarstern* from 1 October 2019 to 16 May 2020. Each of the red circles along the *Polarstern* track

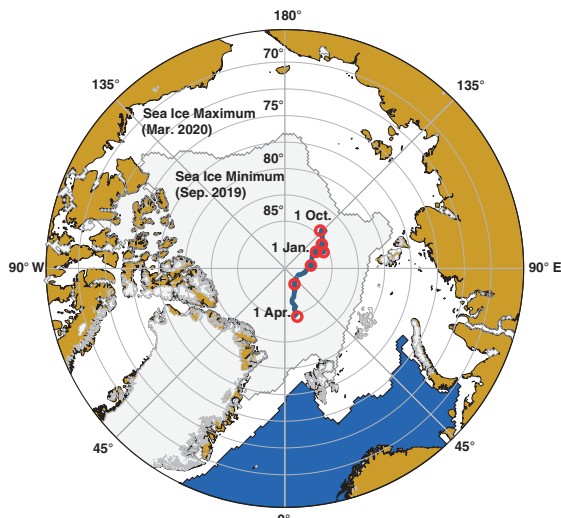

**Figure 1.** Drifting route of the ice breaker *Polarstern* from 1 October 2019 to 1 April 2020. Each of the seven red circles marks the beginning of the next month. The map was produced with 'ggOceanMaps' (Vihtakari, 2020) by using Sea Ice Index Version 3 data (Fetterer et al., 2017).

indicates the beginning of a new month. Most of the time the observations were performed between 85° and 88.5°N during the first six months of the MOSAiC campaign.

The remote sensing instrumentation aboard *Polarstern* mainly consisted of the Atmospheric Radiation Measurement (ARM) mobile facility AMF-1 of the US Department of Energy (http://www.arm.gov, last access: 22 January 2024) and the OCEANET-Atmosphere container of the Leibniz Institute for Tropospheric Research (TROPOS) (Engelmann et al., 2016). These containers were deployed on the bow (front deck) of the *Polarstern*. Photographs of the main ship-based MOSAiC atmospheric measurement platforms aboard *Polarstern* are shown in Fig. 3 in Shupe et al. (2022) and Fig. 2 in Engelmann et al. (2021).

Two lidars transmitting laser beams at 532 nm (visible green light) into the atmosphere were operated continuously aboard *Polarstern* throughout the one year expedition. Figure 2 shows the two beams above *Polarstern*. The picture was taken with a drone overflying *Polarstern* on 31 October 2019.

## 2.1 Polly lidar

The multiwavelength polarization Raman lidar Polly (POrtabLe Lidar sYstem) (Engelmann et al., 2016) performed measurements from 26 September 2019 to 2 October 2020 (Polly, 2024). A detailed description of the Polly instrument can be found in Hofer et al. (2017) and Jimenez et al. (2020). The basic aerosol observations comprise height profiles of the particle backscatter coefficient at 355, 532, and 1064 nm, the particle extinction coefficient at 355 and 532 nm, the respective extinction-to-backscatter ratio (lidar ratio) at 355 and 532 nm, and the particle linear polarization ratio at 355 and 532 nm (Baars et al., 2016; Hofer et al., 2017; Ohneiser et al., 2021). The retrieval of smoke microphysical properties is outlined in Ansmann et al. (2021)

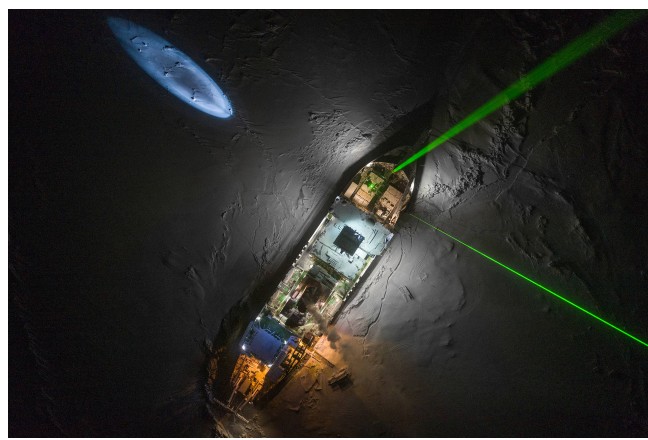

**Figure 2.** Drone-based photograph of *Polarstern*, drifting with the pack ice, along the route shown in Fig. 1. The two green laser beams are produced by the ARM lidar (left beam, exactly vertically pointing) and the TROPOS lidar (right beam, 5° off-zenith pointing to avoid strong specular reflection by falling, horizontally aligned ice crystals). The picture was taken by Jakob Stark and is shown within the Geo journal report of Esther Horvath (https://geo.pageflow.io/gefangen-im-eis#240143, authorship: Alfred-Wegener-Institut / Esther Horvath and Jakob Stark (CC-BY 4.0))

and Ansmann et al. (2023). By means of the measured upper tropospheric smoke backscatter coefficients, the particle surface area concentration (PSAC), the mass concentration, and the number concentrations $n_{50}$ and $n_{250}$ considering particles with radius >50 nm and >250 nm, respectively, can be estimated. PSAC is the aerosol input in the retrieval of the DIN INP number concentration for organic particles (Wang and Knopf, 2011). $n_{250}$ is interpreted as the reservoir of potential INPs. More details to the conversion of optical into microphysical properties is given in Sect. 3.2. The data analysis regarding the microphysical properties of ice crystals is described in Sect. 2.3.

## 2.2 ARM cloud radar

We used the 35 GHz cloud Doppler radar measurements (Ka-band ARM Zenith Radar, KAZR) of the ARM (Atmospheric Radiation Measurement) mobile facility 1 (AMF-1) (ARM, 2024). KAZR measures the radar moments, including reflectivity, mean Doppler velocity, and spectrum width, which provide insight into the mass, size, and fall speed of cloud and precipitation particles (Shupe et al., 2022). Additionally, the vertically pointing radar recorded the full Doppler spectrum, which offers further insight into the cloud particle populations and processes.

## 2.3 Cirrus-related data analysis

The classical Raman lidar technique is used to obtain cirrus optical properties (Ansmann et al., 1992). In the determination of the backscatter coefficient at 532 nm, no assumption of the extinction-to-backscatter ratio (lidar ratio) is required. Furthermore, the solutions are not affected by any multiple scattering effect. After multiplication of the backscatter coefficients with a typical

single-scattering lidar ratio of 32 sr at 532 nm, the desired cirrus extinction coefficient profile is revealed. The 532 nm cirrus lidar ratio is in the range of 28-35 sr (Seifert et al., 2007; Giannakaki et al., 2007; Garnier et al., 2015; Josset et al., 2012; Haarig et al., 2016; Voudouri et al., 2020). Integration from cirrus base to top height yields the 532 nm cirrus optical thickness. To avoid a strong bias in the extinction profiles caused by specular reflection by horizontally oriented falling ice crystals Thomas et al. (1990) the laser beam was directed to an off-zenith angle of 5°. In the case of zenith-pointing lidars the backscatter and extinction coefficients and related cirrus optical thickness can be easily overestimated by a factor of 10. More details to this problem can be found in Mamouri et al. (2023).

An automated (unsupervised) data analysis was applied to the entire MOSAiC lidar data set collected from 1 October 2019 to 31 March 2020. In the first step, we calculated 1 hour mean signal profiles. About 4300 1-hour profiles are theoretically possible within 180 days. We obtained 1716 1-hour profiles showing atmospheric backscatter up to the stratosphere. These profiles were not influenced by low-level clouds and fog and could thus be considered in the cirrus statistics. Further 220 2-hour mean signal profiles up to stratospheric heights could be considered. A longer averaging period was necessary in these cases because of the low signal-to-noise ratio in the case of respective 1-hour profiles. All in all, 1936 lidar profiles were available for further use. 30% of these profiles showed cirrus signatures (652 hours, 587 profiles). The months October, November, and December 2019 contribute with 54, 48, and 124 hours to the 652 cirrus hours. In January, February, and March 2020, we collected 197, 109, and 120 hours of cirrus data.

The following criteria were applied to identify cirrus layers and to determine base and top heights of the cirrus layer. First of all we checked the radiosonde profile of the temperature and restricted the cirrus identification to heights above the $-28°C$ temperature level. Then we used two criteria to identify the cirrus layer. The volume depolarization ratio must exceed coherently (over a vertical range >300 m) the 10% level and, at the same time, the particle backscatter coefficient must exceed 0.1 $Mm^{-1}$ $sr^{-1}$. Identified cirrus layers with a vertical depth of <300 m were thus reject. Since the majority of ice clouds formed in the uppermost part of the troposphere we included the tropopause information from the radiosonde observations in our cirrus studies. The tropopause height was determined following the NASA procedure described in Ohneiser et al. (2021). The results of the cirrus statistical analysis are presented and discussed in Sect. 3.1.

## 2.4 ICNC and IWC retrieval

The retrieval of the ice crystal number concentration ICNC and of the ice water content IWC is based on the synergy of 35 GHz KAZR and 532 nm backscatter lidar observations. The LIRAS-ice (LIdar RAdar Synergy - retrieval of ICE microphysical properties) analysis scheme (Bühl et al., 2019) was originally developed to investigate the impact of Saharan dust on mixed-phase and ice clouds over the Eastern Mediterranean (Ansmann et al., 2019).

LIRAS-ice makes use of the measured profiles of the radar reflectivity factor $Z$ (8.5 mm wavelength) and of the cirrus (single scattering) particle extinction coefficient $E$ at 532 nm wavelength. A careful and accurate determination of the $E$ profile is of fundamental importance for a trustworthy inversion of the combined radar-lidar observations. The optimum, most robust $E$ profiles are obtained by means of the Raman lidar method described in Sect. 2.3.

The applied numerical inversion technique LIRAS-ice (Bühl et al., 2019) is based on a look up table (LUT) which contains the properties of the particle size distribution PSD (assumed to be a monomodal gamma size distribution) (Hogan et al., 2003; Sekelsky et al., 1999; Ulbrich, 1983) and values of $Z$ and $E$. PSD (i.e., the ice crystal number concentration $N(D)$ as function of diameter $D$), $Z$, and $E$ are computed with Eqs. (1), (2), and (5) in Bühl et al. (2019), respectively. PSD is a function of the median particle maximum diameter $D_\mathrm{m}$ and the shape parameters $\mu$ (describing the tilt of the gamma size distribution). More details are given below. In our MOSAiC related modeling effort (LUT computations) we follow Bühl et al. (2019): $Z$ is a function of crystal particle mass (Eq. (B1) with parameters $\alpha = 0.012$ and $\beta = 2.4$ in the case of the MOSAiC data analysis. $E$ depends on the crystal surface area or geometrical cross section (Eq. B2) with parameters $\gamma = 0.17$ and $\sigma = 1.8$). The parameters $\alpha, \beta, \gamma$ and $\sigma$ are given in Tables A1 and A2 in Bühl et al. (2019). For the crystal shape assumption (defined in Table A2) we assume hexagonal plates for the diameter range from 15 to 100 $\mu$m and an aggregate mixture for the sizes from 600 $\mu$m to 5 mm.

The ice crystal diameter range from 100 $\mu$m to 5 mm is considered in the inversion procedure. This assumption reflects realistic characteristics of the crystal size distribution in (aged) ice crystal virga in the case of the synoptic cirrus category (Lynch et al., 2002). The number concentration and size of the crystals in the virga depend on ice crystal growth and collision and aggregation processes. Nucleation of new ice crystals and thus a potential occurrence of a second mode in the size distribution can be ignored at relative humidity over ice of around 100% (i.e., in the absence of strong ice supersaturation) as typically observed with MOSAiC radiosondes in the cirrus virga. The assumption of a monomodal gamma size distribution of the crystals is in agreement with other Arctic cirrus observations (Wolf et al., 2018, 2019; De La Torre Castro et al., 2023). In Sect. 3.5, we will show a comparison with disdrometer-derived ICNC values that corroborate that our approach is fully justified. Our extended sensitivity analysis revealed that the selected size range (e.g., from 10, 25, or 100 $\mu$m up to 5000 $\mu$m) does not play a role in the ICNC retrieval when assuming a monomodal gamma size distribution for the well developed crystal size distributions in ice virga.

In the next step, we estimated the median diameter $D_\mathrm{m}$ of the PSD from $Z$ and $Z/E$ by comparison with simulated cloud radar spectra and lidar parameters (stored in the LUT). A fixed shape parameter of $\mu = 2$ of the gamma size distribution is used (Eq. (1) in Bühl et al. (2019)). Finally, we scaled the results with the observed values of $E$ to obtain ICNC and IWC profiles. The main goal of the retrieval is thus to find a PSD that leads to the same variables as the measured ones ($Z$, $E$). The method developed by Bühl et al. (2019) was well tested and applied to cirrus observations over the Eastern Mediterranean (Ansmann et al., 2019).

We applied the recently published CAPTIVATE (Cloud, Aerosol and Precipitation from mulTiple Instruments using a VAriational TEchnique) algorithm (Mason et al., 2023) to the combined MOSAiC lidar-radar cirrus data sets as well. We found in general good agreement between our LIRAS-ice and the CAPTIVATE results for IWC. However, the ICNC solutions deviate because the ICNC retrieval is rather sensitive to the lidar and radar input data and assumptions on the crystal shape characteristics. CAPTIVATE uses the directly measured attenuated backscatter coefficients (i.e. calibrated range-corrected backscatter signals) as input and derives the required single scattering extinction coefficients as part of the data analysis. Our experience with CAPTIVATE shows that the estimation of the single scattering extinction profile can be a source of significant uncertainty.

**Table 1.** Overview of Polly observational products, used in this study, and typical relative uncertainties in the determined and retrieved properties. $r$ denotes aerosol particle radius.

| Smoke and cirrus properties | Uncertainty |
|---|---|
| Cirrus top height, depth [m] | $< 1\%$ |
| 532 nm cirrus backscatter coef. $[\mathrm{Mm}^{-1}\,\mathrm{sr}^{-1}]$ | $\leq 5\%$ |
| 532 nm cirrus extinction coefficient $[\mathrm{Mm}^{-1}]$ | $\leq 15\%$ |
| 532 nm cirrus optical depth | $\leq 10\%$ |
| Smoke particle surface-area conc. $[\mu\mathrm{m}^2\,\mathrm{cm}^{-3}]$ | $\leq 25\%$ |
| Smoke particle number conc. ($r > 250$ nm) $[\mathrm{cm}^{-3}]$ | $\leq 25\%$ |
| Ice nucleating particle number concentration $[\mathrm{L}^{-1}]$ | Order of magn. |
| Ice crystal number concentration $[\mathrm{L}^{-1}]$ | Order of magn. |

As discussed in Bühl et al. (2019), the uncertainty in the ICNC estimation is roughly characterized by a factor of 3 (around the most reasonable solution). Thus, LIRAS-ice allows us to determined the order of magnitude of occurring ICNC in cirrus fall strikes. This uncertainty margin holds in general for all ICNC radar-lidar retrievals. Table 1 provides an overview of the uncertainties in the observed and retrieved aerosol and cirrus products.

It should mentioned at the end that cloud radar observations up to cirrus top heights were in most cases not possible during the MOSAiC expedition. High quality radar observations could be realized in cirrus virga up to 8 km height, while the top of the cirrus layer was frequently at 9-10 km height. In order to compare simulated and observation-based ICNCs in part 2 (Ansmann et al., 2025), we were thus forced to estimate ICNC values for the main ice nucleation zone in the cirrus top region from the ICNC values available for heights up to 8 km. Aggregation processes can lead to a considerable reduction of the ICNC with decreasing height. ICNCs at 8 km may be a factor of 2-5 lower than respective values in the ice nucleation zone close to cirrus top as a result of crystal-crystal collision and aggregation events. Aggregation effects are discussed in Sect. 3.3.

## 2.5 Two-dimensional video disdrometer (2DVD)

The two-dimensional video disdrometer (2DVD) is a ground-based precipitation gauge which detects single precipitation particles within a certain measuring area (Kruger and Krajewski, 2002; Gaudek, 2024). The 2DVD was originally designed to measure rain drop size distributions. The investigation of solid hydrometeors with such devices has rather been subject of research in recent years. The instrument was developed by Joanneum Research, Graz, Austria (https://www.joanneum.at/). During the MOSAiC expedition it was operated on the roof of the TROPOS OCEANET-Atmosphere container. The 2DVD, including a product characterization, is described in detail in Gaudek (2024).

The 2DVD allowed us to measure ICNC on a calm day (23 November 2019) and to compare the values with respective numbers from the lidar-radar retrieval. Furthermore, the 2DVD observations provided information on the ice crystal sizes,

terminal velocities of ice crystals, and whether the ice crystals were compact or rather complex in shape and thus hints on the importance of crystal-crystal collision and aggregation processes.

## 2.6 *Polarstern* radiosonde

Vaisala radiosondes (type RS41) were launched regularly every 6 hours throughout the entire duration of MOSAiC, including periods when *Polarstern* was in transit (Maturilli et al., 2021, 2022). The radiosondes provide vertical profiles of temperature, relative humidity, pressure, and winds from 12 m (the altitude of the helideck from which they were launched) up to an altitude of about 30 km, thus covering both troposphere and lower stratosphere (Shupe et al., 2022). Quality control for appropriate physical ranges has been applied.

## 3 Observations

The MOSAiC observations and retrieval products are presented and discussed in several subsections. In Sect. 3.1, we begin with the results of a statistical analysis of observed cirrus geometrical and optical properties. In Sect. 3.2, we provide an overview of the MOSAiC aerosol pollution conditions at tropopause level where ice formation usually started. In Sect. 3.3 – 3.5, we present our findings regarding the microphysical properties of 12 cirrus systems (20 profile data sets) observed in the winter months (November 2019 to February 2020) and discuss the potential impact of smoke on the evolution of the observed cirrus systems. This also includes a discussion on the importance of observations of ice crystal number concentrations (ICNCs) in the cirrus virga. These virga ICNC observations connect part 1 and the simulations in part 2, as will be explained in Sect. 3.4.

## 3.1 Cirrus statistics for the winter half year 2019-2020

Figure 3 presents the statistical results of the cirrus observations from 1 October 2019 to 31 March 2020. Although the observations cover the winter half year only the comparison with other studies (discussed below) suggest that they are representative for the entire year. During the summer half year, less than 20 cirrus profiles (one hour mean profiles) could be collected. Low clouds and fog prohibited upper tropospheric measurements most of the time during spring and summer months. According to the cirrus classification of Lynch et al. (2002) all of the observed Arctic winter cirrus clouds belong to the synoptic cirrus category (top-down generation of cirrus structures). Ice nucleation starts at cloud top, where usually the highest values of the ice saturation ratio are observed and, later on, extended virga of falling ice crystals evolve, and reach lower and lower heights. An example is discussed in the next section. De La Torre Castro et al. (2023) reported that 86% of the cirrus layers they observed during a summer campaign in June and July 2021 at latitudes from 60-76°N, belonged to the synoptic cirrus category. 14% of the ice clouds were orographically induced cirrus or anvil cirrus.

As outlined in Sect. 2.3, the MOSAiC cirrus statistics in Fig. 3 is based on 1716 1-hour mean and 220 2-hour mean signal profiles. They cover 2156 hours and thus almost 50% of the theoretically possible 4320 1-hour profiles of a half year. All profiles showing cirrus signatures and cloud top temperatures $< -28°C$ are considered in the statistics. In more than 90% out of all cases the cirrus top temperature was $< -40°C$.

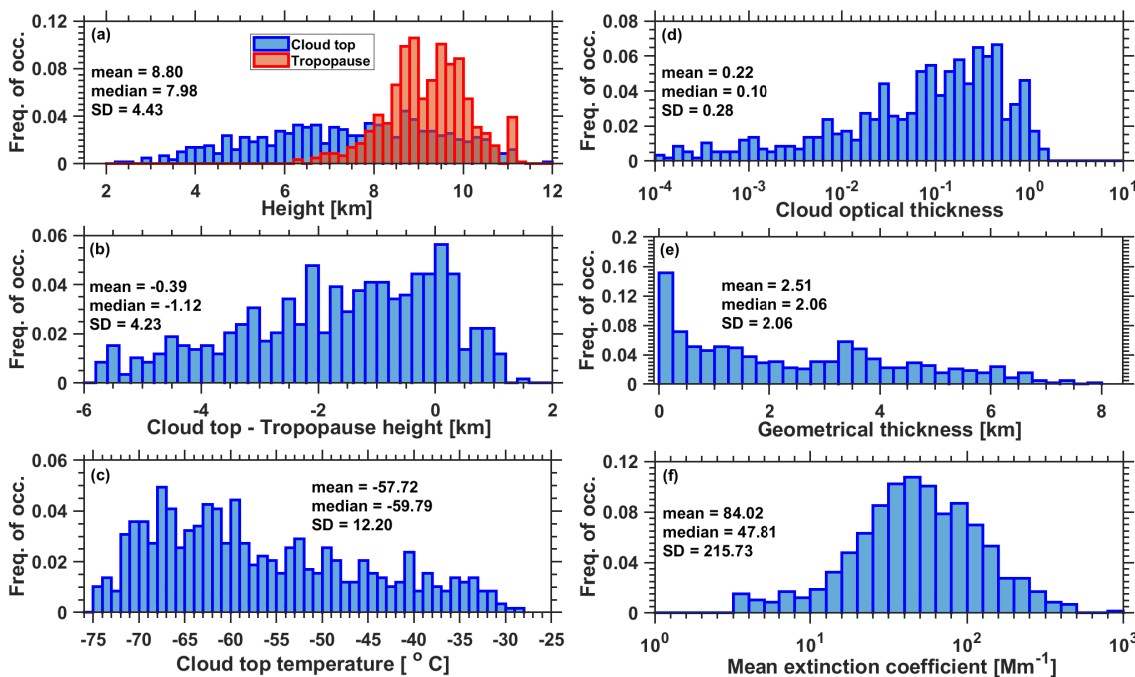

**Figure 3.** MOSAiC cirrus statistics considering all available lidar observations from 1 October 2019 to 31 March 2020. The normalized frequency of occurrence is shown. (a) Cirrus top height and tropopause, (b) Vertical distance of the cirrus top height from the tropopause, (c) cirrus top temperature, (d) cirrus optical thickness (532 nm), (e) cirrus vertical extent (from cirrus top to virga base height), and (f) cirrus mean extinction coefficient at 532 nm. Median, mean,and SD values are given as numbers.

The main findings can be summarized as follows: More than 30% out of all 1936 profiles showed cirrus signatures (652 hours, 587 profiles). For comparison, typical cirrus occurrence frequency was of the order of <10% according the space lidar observation with CALIOP (Cloud-aerosol lidar with orthogonal polarization) at latitudes around 80°N in 2006-2007 (Sassen
245 et al., 2009; Heymsfield et al., 2017). Most cirrus clouds developed at or close to the tropopause (Fig. 3a,b). The tropopause height distribution, derived from the radiosonde observations, is shown in Fig. 3a. Many detected cirrus features far below the tropopause are related to virga fragments (Fig. 3b). Most ice nucleation occurred at temperatures from −60 to −75°C or 198-213 K (Fig. 3c, radiosonde observations). The values for the cirrus optical thickness accumulate in the range from 0.05 to 0.5 (Fig. 3d). Sassen and Cho (1992) classified cirrus clouds as subvisible cirrus when the cirrus optical thickness COT is <0.03,
250 as visible cirrus when the COT ranges from 0.03-0.3, and as opaque cirrus when COT>0.3. According to this classification, 25% of the observed central Arctic cirrus clouds were subvisible, 40% visible, and 35% opaque cirrus. The broad distribution of the cirrus vertical depth in Fig. 3e indicates the impact of the strongly varying virga base height, interpreted as the cirrus base height. The cirrus mean extinction coefficient in Fig. 3f shows a Gaussian distribution (in logarithmic scales) with typical extinction values from 30 to 300 Mm$^{-1}$ (or 0.03-0.3 km$^{-1}$.).

Our statistical results are in good agreement with other studies of Arctic cirrus properties. Heymsfield et al. (2017) stated that typical cirrus top temperatures are between 200 and 213 K and the cirrus top height ranges from 8-14 km in the Arctic. Schäfer et al. (2022) analyzed ground-based lidar observations at the ALOMAR site in northern Norway (69.1°N) and compared the findings with respective results from CALIOP overflights (2011-2017). Typical cirrus top temperatures were in the range of 210-220 K (fall, winter) and 220-230 K (spring, summer). Cirrus top was mostly at heights from 8.5-10.5 km. Cirrus base heights (i.e., in most cases virga base heights) ranged from 4-11 km over the ALOMAR lidar site. Nakoudi et al. (2021) analyzed lidar data collected over Ny Ålesund, Svalbard, Norway (78.6°N) from 2011-2020, and also found cirrus top heights mainly between 8.5 and 10.5 km (throughout the year) and cirrus top temperatures accumulated between 203 and 213 K. Voudouri et al. (2020) analyzed long-term observations (2011-2016) in Finland (Kuopio, 62.7°N). Cirrus top heights were mostly between 9 and 10 km in winter and 10-10.5 km in summer. Mean COT at 532 nm was found to be 0.24±0.2, with 3% contributing to the subvisible cirrus fraction, 71% to the visible cirrus fraction, and 26% to the opaque cirrus fraction. These numbers are very different from the MOSAiC values of 25% subvisible, 40% visible, and 35% opaque ice clouds. Orographically forced waves generated by the Scandinavian mountains have probably a strong impact on the cirrus characteristics over Finland. It should be mentioned at the end, that the statistical results in Fig. 3 do not provide any hint regarding the dominating ice nucleation mode (homogeneous vs heterogeneous ice nucleation).

## 3.2 Upper tropospheric aerosol conditions during the winter 2019-2020

During the MOSAiC winter half year the UTLS in the central Arctic was covered by a thick layer of aged Siberian wildfire smoke (Ohneiser et al., 2021; Ansmann et al., 2023, 2024). In terms of optical properties, the MOSAiC smoke 532 nm extinction coefficients in the UTLS height range were with 3-5 Mm$^{-1}$ about a factor of 20 higher than respective extinction coefficients for the UTLS background sulfate aerosol of 0.1-0.25 Mm$^{-1}$ (Jäger, 2005; Baars et al., 2019). The measured smoke extinction coefficients were used to estimate ice-nucleation-relevant quantities such as the particle number concentrations $n_{250}$ (considering particles with radius $r > 250$ nm) and the particle surface area concentration (PSAC) as described in Ansmann et al. (2021) and also in Ansmann et al. (2025).

In Fig. 4, the PSAC time series at tropopause level from October 2019 to March 2020 is shown. The PSAC values are used as input in the INP parameterization within the simulation studies in part 2 (Ansmann et al., 2025). The corresponding particle number concentration $n_{250}$ can be interpreted as an INP reservoir, containing all particles that can eventually be activated as INPs (Knopf et al., 2023). The PSAC values of 10 $\mu$m$^2$ cm$^{-3}$ corresponds to $n_{250}$ values of about 2000 L$^{-1}$. According to Schröder et al. (2002) the liquid background sulfate particle number concentration, i.e., the INP reservoir in the case of homogeneous freezing processes, is of the order of 250 cm$^{-3}$ (or 250000 L$^{-1}$) in the upper troposphere. The respective particle volume concentration of the background aerosol is around 1 $\mu$m$^3$ cm$^{-3}$. The particle volume concentration serves as aerosol input in homogeneous ice nucleation computations (Koop et al., 2000).

As can be seen in Fig. 4, the smoke layer was continuously observed since the beginning of the MOSAiC expedition and the pollution level did not change from December 2019 to March 2020. The tropopause temperatures decreased slowly from November 2019 to March 2020. The PSAC values accumulated in the range of 5-15 $\mu$m$^2$ cm$^{-3}$. A rather strong and long-living

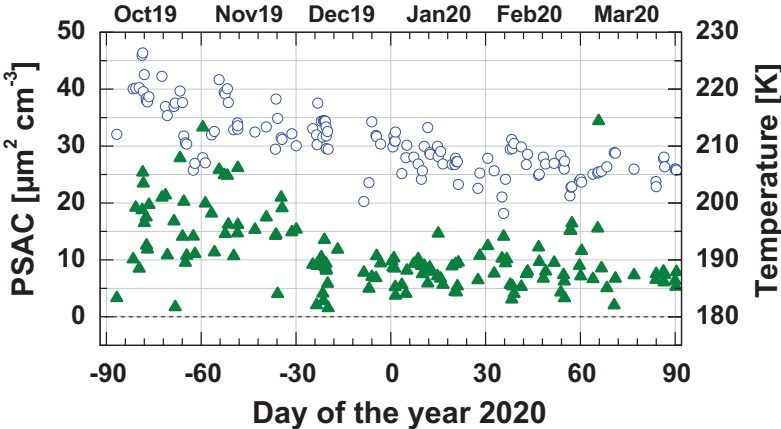

**Figure 4.** Smoke particle surface area concentration (PSAC) at tropopause height level (green triangles) together with the tropopause temperature (open blue circles) from the beginning of October 2019 to the beginning of April 2020. The lidar observations of PSAC and the radiosonde temperature observations were performed during cirrus-free periods.

polar vortex controlled the weather and vertical and horizontal aerosol transport conditions from December 2019 to April 2020
(Ohneiser et al., 2021). As a consequence, almost constant smoke particle concentration levels were observed during this time period. A depletion of the smoke particle reservoir by frequently occurring cirrus formation events is not visible. The INP reservoir in the upper troposphere was permanently refilled from above. According to Ohneiser et al. (2021) and Ansmann et al. (2024), the smoke layer extended from 5-7 km height up to about 12-13 km height, and thus up to several kilometers above the tropopause.

The PSAC observations in Fig. 4 were conducted during cirrus-free periods to avoid contamination of the aerosol backscattering by cirrus backscattering. We assume that the observed PSAC values also describe the pollution conditions during cirrus ice nucleation processes at cirrus top heights. An impressive example of cirrus formation in the smoke-polluted tropopause region was observed from 25-29 February 2020, shown in Fig. 14 in Ansmann et al. (2023).

### 3.3  22 January 2020 cirrus case study

A mid-winter MOSAiC cirrus case study is presented in Figs. 5-8. The measurement was performed on 21-22 January 2020 at 87.5°N. From 21 January, 12 UTC, to 26 January, 20 UTC, and thus over six days, cirrus and virga formation continuously occurred over *Polarstern*. The evolution of two cirrus systems are highlighted in Fig. 5. Ice nucleation was probably initiated mainly at heights >10 km and at temperatures of −70 to −74°C. Nucleated ice particles grow fast by water vapor deposition on the available crystal surfaces. The respective reduction of the ice saturation ratio $S_i$ below the ice nucleation onset value $S_{i,on}$ terminates the ice nucleation event (as will be shown in part 2). The growing and falling crystals form vertically extended and coherent virga structures. According to Bailey and Hallett (2004, 2012) crystals grow by 0.01-0.05 $\mu$m s$^{-1}$ (diameter growth per second) at temperatures from −50 to −30°C (3-7 km height range in Fig. 5) and ice supersaturation levels around

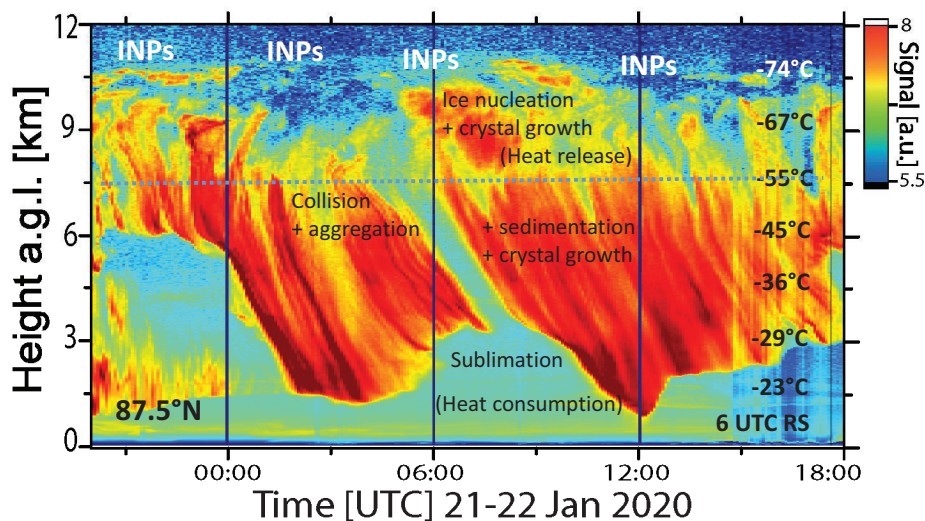

**Figure 5.** Life cycles of two cirrus systems, the first one from about 21 UTC on 21 January to 7 UTC on 22 January, and the second one from 6 UTC to 16 UTC on 22 January 2020. Cirrus structures are mainly given in yellow and red colors. The *Polarstern* lidar observation (in terms of the calibrated range-corrected 1064 nm backscatter signal) was performed at 87.5°N. The cirrus clouds belong to the synoptic cirrus category (top down generation of cirrus structures). Cirrus generation cells were mainly above 9 km height, the virga zone extended from about 9 km down to 1 km above *Polarstern*. Temperatures (given as numbers on the right side) were measured with 6 UTC radiosonde. The vertical black lines indicate 0, 6, and 12 UTC radiosondes (launched at 23, 5, and 11 UTC, respectively). All relevant processes are indicated such as nucleation, growth, sedimentation, and aggregation. The blue dotted line shows the maximum height up to which cloud radar reflectivity was available to retrieve microphysical properties. Smoke particle INPs showed a maximum at the tropopause level around 10.5-11 km height.

1.1. In the case of the virga observed between 1.5 and 7.5 km height from 8-12 UTC, crystals probably grew to sizes of 150-750 $\mu$m in these four hours (14400 s). Different crystal nucleation times and different growth rates at different heights and temperatures (Bailey and Hallett, 2004, 2012) lead to a broad spectrum of ice crystal sizes. The resulting fall speed spectrum may foster crystal-crystal collision and subsequent aggregation processes and may lead to a considerable number of crystals with diameters exceeding even 1 mm in the lower half of the virga height range before sublimation of the crystals in the lowest part of the virga start to dominate. The strong increase of the lidar backscatter signal strength with decreasing height in Fig. 5 is a clear sign for the increasing size of ice crystals. The ice particles moved downward from heights of 7.5 km to 1.5 km within the time period from 8-12 UTC and thus with an apparent mean falling speed of about 40 cm s$^{-1}$. The Doppler radar observations indicated falling velocities around 50 cm s$^{-1}$.

According to the HYSPLIT backward trajectories in Fig. 6 the humid air mass, in which the cirrus formed, originated from the Pacific Ocean. The air mass spent 4-5 days in the polluted upper troposphere over the Arctic before first cirrus were detected above *Polarstern* on 21 January 2020. During these 4-5 days there was sufficient time for entrainment of aged wildfire smoke particles into the moist air from above, i.e, from the main smoke reservoir in the lower stratosphere (Ohneiser et al., 2021).

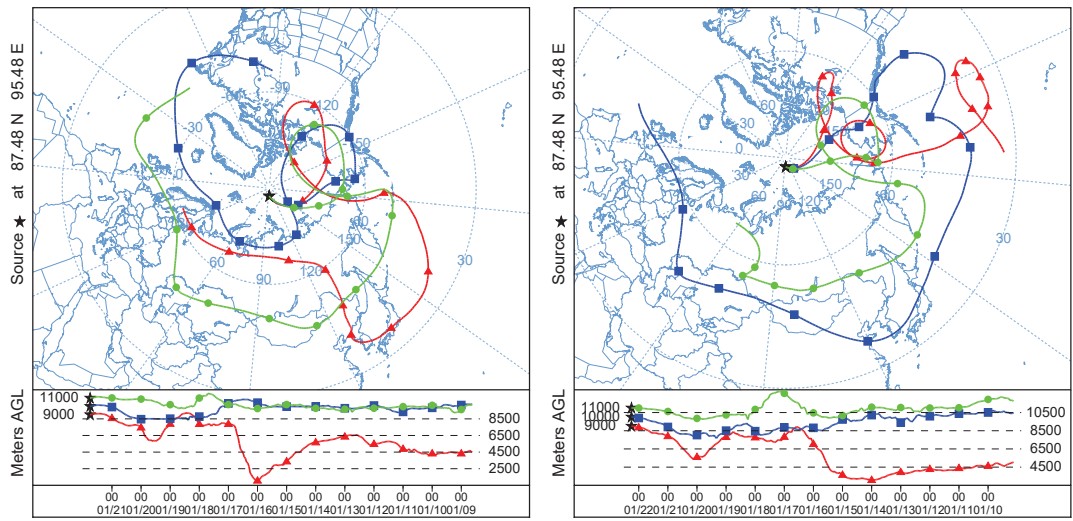

**Figure 6.** HYSPLIT 10 day backward trajectories, arriving over *Polarstern* (indicated by a star) on (a) 21 January 2020, 18:00 UTC and (b) on 22 January 2020, 6:00 UTC (HYSPLIT, 2024; Stein et al., 2015; Rolph et al., 2017). Arrival heights are at 9000 m (red), 10000 m (blue), and 11000 m (green).

As mentioned in Sect. 3.2, the aerosol observations suggest that the smoke INP reservoir was permanently refilled from above during the cirrus evolution processes on 21-22 January 2019. This is highlighted in Fig. 5 (see 'INPs' written in the figure above and within the cirrus top region).

As can be seen in Fig. 5, the lidar is able to detect any cirrus structure from the top of the cirrus (ice generation cells) at 10.5 km height to the base of the virga. In contrast, the cloud radar reflectivity was useful for further analysis up to heights of 7.5 km, only. The blue dashed line in Fig. 5 marks the height up to which radar reflectivity data were available for the derivation of cirrus microphysical properties.

The microphysical properties in the virga height range in Fig. 7 were derived from the combined lidar and radar observations by means of our LIRAS-ice analysis scheme (Sect. 2.4). The top panel shows the $Z/E$ data field that served as measured input

in the retrieval. The derived IWC mostly shows values from 0.0001 to 0.01 g m$^{-3}$ and the ICNC ranged from 0.1 to about 50 L$^{-1}$. The analysis of the lidar-radar observations, performed from 17-24 UTC, became difficult and is less trustworthy. The vertical white columns and the white areas Fig. 7 indicate fields without retrieval products. In addition, the results at the boundaries of analyzed data fields must be interpreted with caution because lidar and radar do not see exactly the same air volumes so that $Z$ and $E$ can frequently no longer be combined without introducing significant uncertainties in the products.

Many ice virga show up as pronounced coherent structures in the height-time displays of IWC and ICNC in Figs. 7b and 7c. We assume that each well-resolved virga is linked to a singular, individual ice nucleation event so that the virga occurrence frequency is equal or almost equal to occurrence frequency of updraft and ice nucleation events. The retrieved ICNC values and the observed temporal width and structural complexity of the virga contain information about updraft strength, duration,

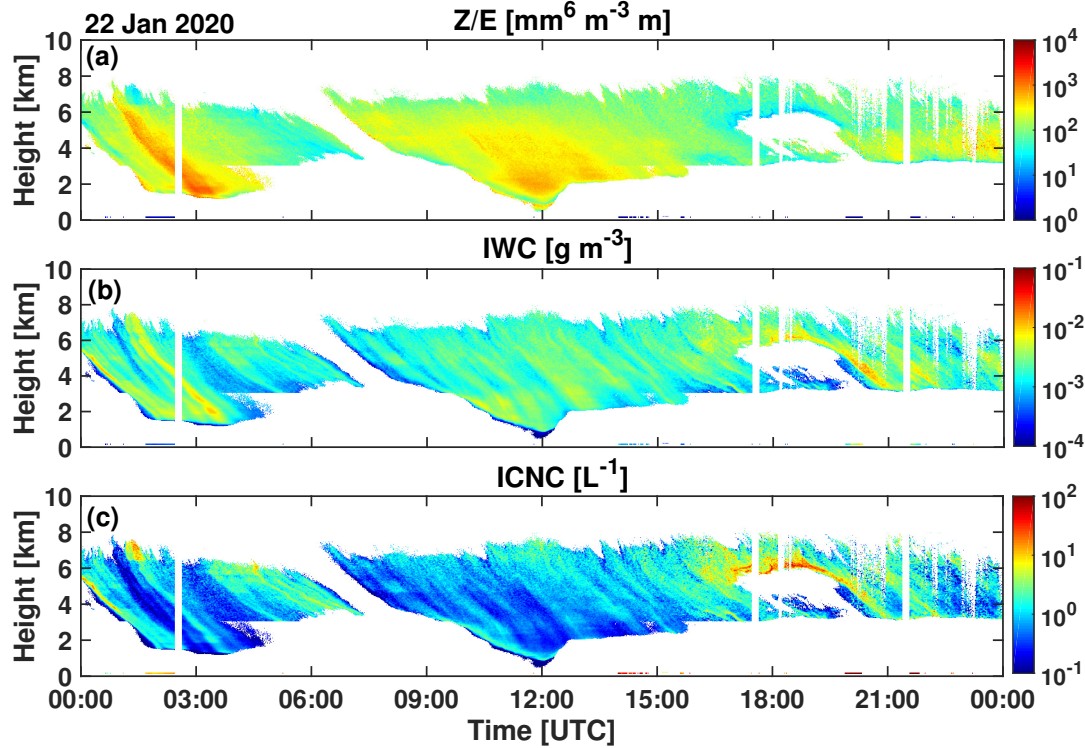

**Figure 7.** (a) Ratio $Z/E$ with radar reflectivity $Z$ and 532 nm cirrus extinction coefficient $E$ observed on 22 January 2020, (b) LIRAS-ice retrieval of ice water content IWC, and (c) LIRAS-ice retrieval of ice crystal number concentration ICNC.

and complexity (from single events to complex clusters of updrafts) and respective ice nucleation intensity. The observations
suggest that the virga occurrence frequency is directly linked to the occurrence frequency of updraft and ice nucleation events. The short-term updraft periods during which the nucleation events develop occur randomly and are omnipresent in the upper troposphere as a result of gravity wave activity (Podglajen et al., 2016), windshear-induced turbulence production, and orographic influences. More details to the origin of updrafts and their role in ice nucleation processes are given in Sects. 3.4 and 3.5 and in part 2 (Ansmann et al., 2025).

The temporal width of the virga in Fig. 7 ranged from about 10-20 minutes, sometimes up to 30 minutes. The corresponding horizontal extent was about 3-10 km when considering the radiosonde observations of wind speeds of around 5 m s$^{-1}$ in the height range from 4-8 km height on that day. In well-resolved virga, ICNC ranged from 1-10 L$^{-1}$ in most cases, but sometimes also values up to about 50 L$^{-1}$ are visible. When considering that the measured INP reservoir in the upper troposphere contained about 2000 potential smoke INPs per liter and was permanently refilled from above according to the discussion in
Sect. 3.2, even numerous short-term lofting events, experienced by a given smoke-filled air parcel, will not lead to an empty or almost depleted INP reservoir so that favorable conditions for homogeneous freezing were given and freezing of the liquid background aerosol particles could start and dominate ice nucleation. This aspect is further discussed in the simulation study in

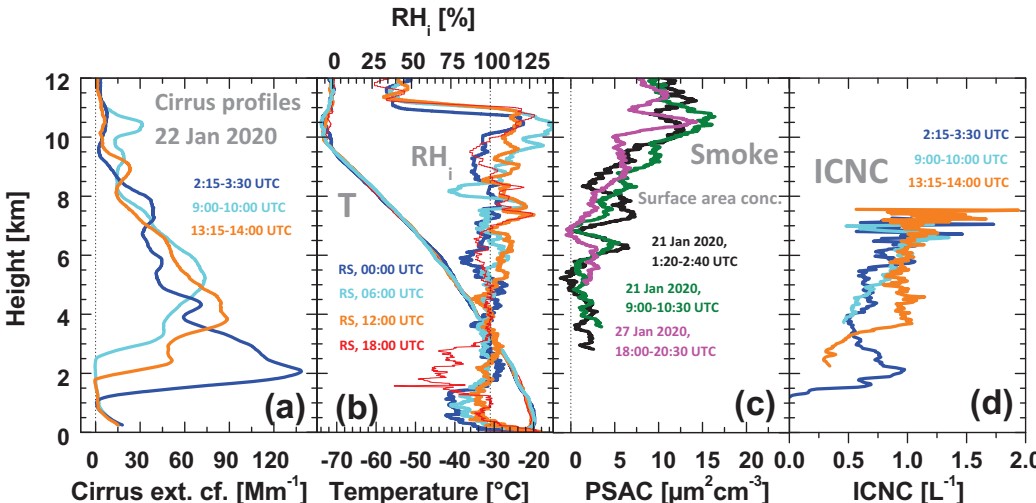

**Figure 8.** Synergistic overview of (a) cirrus geometrical and optical properties, (b) meteorological conditions in terms of temperature $T$ and relative humidity over ice $RH_i$ measured with 4 radiosondes (RS), (c) smoke pollution levels in terms of PSAC, and (d) ice crystal number concentrations (ICNC) in the ice virga, obtained from combined lidar-radar observations.

part 2 (Ansmann et al., 2025). We may therefore conclude that heterogeneous ice nucleation on the smoke particles in the upper troposphere was widely responsible for the observed cirrus fields. However, even in the case of a highly polluted UTLS we

cannot completely rule out that occasionally situations occurred in which almost smoke-free conditions were given in a number of air parcels after the consumption of all or most of the smoke INPs so that homogeneous freezing became an additional ice nucleation option. As shown by Rolf et al. (2012), Krämer et al. (2016) and Kärcher et al. (2022), homogeneous ice nucleation is possible in the presence of a low number of INPs when the updrafts are strong enough so that the diffusional growth of the few heterogeneously nucleated ice crystals by water vapor deposition is not sufficient to reduce the relative humidity in the

ascending air parcel significantly and to prevent that the ice saturation ratio $S_i$ in the lofted air parcel reaches and exceeds the onset ice saturation ratio $S_{i,on}$ for homogeneous freezing.

Figure 8, finally, compiles measured cirrus geometrical and optical properties, meteorological conditions, cirrus relevant smoke properties, and ICNC values. Panel a in Fig. 8 shows mean height profiles (1-2.5 hour mean profiles) of the cirrus extinction coefficient indicating the cirrus height range from top to bottom. Three different periods of cirrus evolution on

22 January (2-14 UTC) are selected. Panel b indicates rather constant meteorological conditions in terms of temperature and relative humidity over ice ($RH_i$) as measured with radiosondes launched at 23 UTC on 21 January, and at 5, 11, and 17 UTC on 22 January. The ice saturation ratio $S_i$ varied around 1.0 (equilibrium conditions, $RH_i = 100\%$ in the more than 6-8 km deep virga height range and showed values up to 1.2-1.35 at the top of the cirrus system (in the ice nucleation zone).

The high $S_i$ value of 1.35 at cirrus top observed with the 6 UTC radiosonde in the beginning of the evolution of a new cirrus

complex, represents a frequently observed MOSAiC cirrus-top ice saturation ratio. This value is significantly lower than the

values of $S_i > 1.47$ reported by Dekoutsidis et al. (2024) for the Arctic cirrus top region. High $S_i$ values above 1.47 indicate the dominance of homogeneous freezing, whereas values around 1.35 point to the dominance of heterogeneous ice nucleation by inefficient INPs as expected when smoke particles (organic particles) serve as INPs. $S_i$ values $> 1.25$ would probably not be observable if efficient INPs such as mineral dust particles are present in the cirrus top region (Ansmann et al., 2019; Dekoutsidis et al., 2024).

Panel c shows profiles of the particle surface area concentration (PSAC) of the smoke particles as estimated from lidar observations during clear-sky conditions in the morning of 21 January 2020 and on 27 January 2020 after the long-lasting 6-day period with strong cirrus developments. The PSAC maximum was found around and just above the tropopause, as always during the winter months from November 2019 to March 2020 (Ohneiser et al., 2021). The PSAC profiles, observed before and after the period with strong cirrus formation from 21-26 January 2020 (further discussed in part 2), do not indicate any decrease in the PSAC values. As mentioned already, we assume that the upper tropospheric smoke INP reservoir was permanently refilled by a downward flux of particles from the lower stratosphere to the upper troposphere. Panel d finally presents mean ICNC profiles (mean values for approximately 1 hour) for the virga height range from 7.5 km down to about 1.5 km height. The retrieved mean ICNC values are mostly between 0.5 and 1 $L^{-1}$. around 1 $L^{-1}$. Note that the hourly mean profiles cover both virga events as well as virga-free time periods. The obtained hourly mean ICNC numbers are thus lower than the ones in the pronounced virga in Fig. 7.

### 3.4 ICNC observations in virga: the link to the simulations in part 2

Figure 9 shows further ICNC height time displays for cirrus events observed on 23 November 2019, 6 December 2019, and 24 January 2020. The 6 December case was partly discussed already in Engelmann et al. (2021). Typical ICNC values in the virga ranged again from 1 to 10 $L^{-1}$. Also virga pattern with ICNC values from 10 to 50 $L^{-1}$ occurred. Radar observations almost up to cirrus top (at 8.5 km height) were possible on 6 December so that ICNC values up to the ice nucleation zone could be retrieved on this day. As mentioned in the foregoing section, the ICNC color displays suggest that the observed virga structures and virga ICNCs contain information about the strength of updraft events (lofting amplitude, lofting duration) and the corresponding ice nucleation intensity in the cirrus generation cells close to cloud top. The virga ICNCs were used as a guide in the development of our simulation strategy in part 2 and as orientation in the design of realistic simulation scenarios.

Figure 9 also provides an impression about a possible impact of crystal-crystal collisions and aggregation events on the ICNC height dependence. In many cases, ICNC decreases with decreasing height within the virga. Besides aggregation effects also sublimation of crystals contribute to the reduction of ICNC with decreasing height. According to Kienast-Sjögren et al. (2013), aggregation processes are of minor importance at $-60$ to $-75°C$. Furthermore, aggregation effects seem to be also of low importance when ICNC $<10 L^{-1}$. Wolf et al. (2018) showed balloonborne ICNC profile observations in an Arctic cirrus deck performed in 12 February 2016. The cirrus vertical structures were similar to the ones measured on 22 January 2020, discussed in the foregoing section. The ICNC values ranged from 1-10 $L^{-1}$ in the height range from 6 to 11 km height on 12 February 2016. A decrease of ICNC from cirrus top to base was not visible in the shown data. In contrast, studies of Field and Heymsfield (2003) and Mitchell et al. (2018) indicate that an underestimation of the ICNC at cirrus top from ICNC values

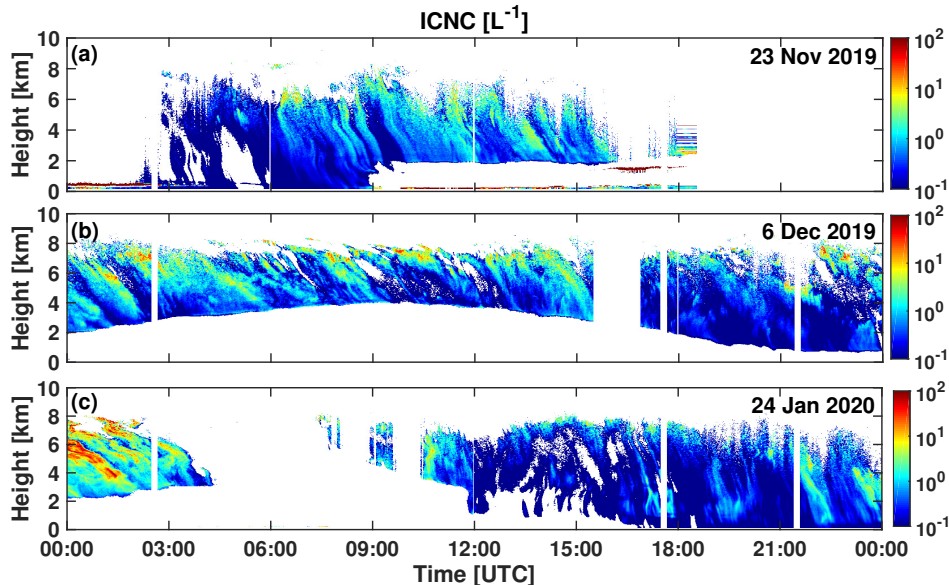

**Figure 9.** ICNC height time displays obtained by means of the LIRAS-ice retrieval scheme in the virga zones for (a) 23 November 2019 (at 85.7°N, 120°E), (b) 6 December 2019 (at 86.1°N, 122°E), and (c) 24 January 2020 (at 87.4°N, 93°E). Cirrus top height was at about 9 km on 23 November (a), 8.5 km on 6 December (b), and 10 km on 24 January (c).

in the upper part of the main observable virga zone by a factor of 2 (moderate impact) or even by a factor of 5 (strong impact) may occur and should be considered in the interpretation of ICNC profiles. Field and Heymsfield (2003) studied midlatitude cirrus with ICNC of the order of 200 $L^1$ in the nucleation zone.

In the case of the MOSAiC observations we assume that the virga ICNC values at 7.5-8 km height can be well used as ICNC estimates for the ice nucleation zones as long as the retrieved ICNC are $<10\ L^{-1}$ and that the cirrus top ICNCs may be a factor of 2 higher than the observed virga ICNC values when the virga values are $>50\ L^{-1}$.

### 3.4.1 ICNC comparison: 2DVD observations vs lidar-radar retrievals

To check the overall quality of our virga ICNC retrievals, we used the opportunity of simultaneous 2DVD, lidar, and radar observations of ice virga reaching the ground on 23 November 2019 (Fig. 10). On this day, favorable calm and almost windless conditions were given with wind speeds of 0-3 m s$^{-1}$ in the lowermost 400 m of the atmosphere according to the radiosondes launched at 5 and 11 UTC. The cirrus virga reached the ground (Fig. 9a) so that the 2DVD aboard *Polarstern* could measure the incoming ice crystal flux. As can be seen in Fig. 10, the ice saturation ratio was $>1.0$ down to the ground at 5-6 UTC so that sublimation of ice crystals during falling is widely suppressed at that time. The temperature at ground was around $-20$°C.

In Fig. 10, we compare 1 hour mean disdrometer values with 2 hour mean ICNC profiles. The agreement between the in situ observations and the remote sensing products is reasonably good when keeping an uncertainty of a factor 3 in the LIRAS-ice products into consideration. Both approaches show low ICNC values $<1\ L^{-1}$ near the ground. The lowest remote sensing

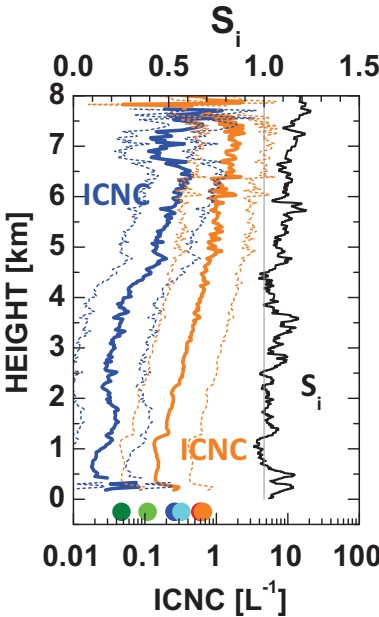

**Figure 10.** Comparison of hourly mean values of ICNC measured with 2DVD aboard *Polarstern* (circles at the bottom of the panel, blue: 3-4 UTC, cyan: 4-5 UTC, olive 5-6 UTC, green: 6-7 UTC, red: 7-8 UTC, orange: 8-9 UTC) with ICNC profiles retrieved from the lidar-radar observations (2 hour mean profiles, 3-5 UTC in blue, 7-9 UTC in orange). The dashed curves show the uncertainty margin (mean profile multiplied and divided by 3). The ice saturation ratio $S_i$ measured with radiosonde launched at 5 UTC is shown in black. The $S_i$ profile indicates that evaporation of ice crystals did not occur over the entire sedimentation height range.

height bin is 250 m above *Polarstern*. The decrease of ICNC with decreasing height mainly reflects the increasing influence of sublimation of ice crystals in the inhomogeneous virga fields below 4-5 km height, especially during the time period from 3-5 UTC (see Fig. 9a), before the 6 UTC sonde was launched at 5 UTC showing ice saturation ratios close to 1.0. Time periods without pronounced virga structures and thus with background-like conditions increased with decreasing height (below 5 km height), probably also the result of an increasing influence of sublimation processes with decreasing height. A significant aggregation impact is not very likely. The ICNC values in the uppermost part of the virga in Fig. 9a were between 0.1 and 10 L$^{-1}$ and the respective 2 hour mean values between 0.1 and 2 L$^{-1}$ in Fig. 10.

The 2DVD observations indicated crystal diameters clearly larger than 150 micrometer, most crystals showed sizes around 500 $\mu$m, and bullet-rosette-like crystal shapes. Assuming a sedimentation velocity of around 50 cm s$^{-1}$, the ice particle reached the ground after about 18000 s when starting from the ice nucleation zone at 9 km height. The crystals could permanently grow over 5 hours by water vapor uptake.

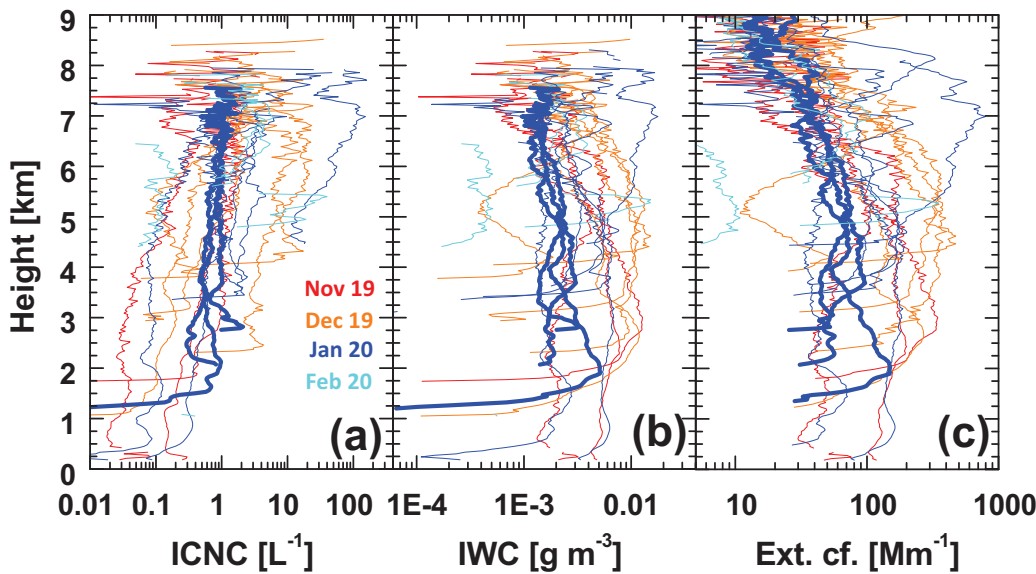

**Figure 11.** Height profiles of (a) ice crystal number concentration ICNC, (b) ice water content IWC, and (c) cirrus extinction coefficient (532 nm). 20 cirrus profiles are shown (mean profiles average over 1-2 hours). November 2019 profiles are given in red, December 2019 profiles in orange, January 2020 profiles in blue, and February 2020 profiles in cyan. The 3 profiles measured on 22 January (see Fig. 8) are shown in thick blue.

### 3.5 Microphyscial properties of Arctic winter cirrus clouds: overview

Figure 11 provides an overview of the microphysical properties of 12 individual cirrus events (1 in November, 2 in December, 5 in January, and 4 in February). Twenty data sets of lidar and radar observations (averaged over 1-2 hours), performed during
435    the 12 cirrus events, were selected and analyzed. The respective 20 profiles of the ICNC, IWC and cirrus extinction coefficient are shown in Fig. 11. Note again that the 1-2 hour mean profiles cover virga and as well as virga-free time periods. The values are thus lower than pure virga values, discussed before.

The ICNC values in Fig. 11 cover a wide range from 0.1 to more than 100 $L^{-1}$ in the upper part of the retrieval height range. Most values were, however, between 0.1 and 10 $L^{-1}$. The IWC showed values from 0.001 to 0.02 g $m^{-3}$ and the cirrus
440    extinction values accumulated between 30 and 300 $Mm^{-1}$, in good agreement with the statistical results shown in Fig. 3f in Sect. 3.1.

Rather similar values as shown in Fig. 11 were obtained during an airborne field campaign at high northern latitudes (60°-76°N, between Scandinavia and Greenland) in June and July 2021 (De La Torre Castro et al., 2023). The airborne in situ observations revealed median values of 1 $L^{-1}$ (0.01-9 $L^{-1}$, 25th to 75th percentile) for ICNC, and 0.0019 g $m^{-3}$ (0.0002 to
445    0.011 g $m^{-3}$, 25th to 75th percentile) for IWC. Cirrus extinction coefficients accumulated between 1 and 500 $Mm^{-1}$.

At the end of our MOSAiC cirrus data analysis, the question remains: Why were the ICNC values in the central Arctic cirrus clouds, on average, so low? We hypothesize that the amplitudes of the omnipresent updrafts may have an important impact on the ICNC levels. The updraft amplitude defines the height range available for an air parcel to ascent. If the amplitude is sufficiently large so that the steadily increasing ice saturation ratio $S_i$ in the rising air parcel can reach and exceed the ice nucleation onset value $S_{i,on}$, ice crystals can form. The lower the amplitude, the lower is the remaining lofting range for ice nucleation and therefore the lower is the total number of ice crystals formed during an updraft event. More details to the dependence of ICNC on the updraft amplitude is given in part 2 (Ansmann et al., 2025).

Podglajen et al. (2016) quantified wave-induced Lagrangian fluctuations of temperature, vertical displacement of air parcels, and vertical velocity in the lower stratosphere (in the 15-18 km height range) over polar regions by using measurements with superpressure balloons (SPBs). Observations recorded every minute along SPB flights allowed the whole gravity wave spectrum (up- and downdraft events) to be described and provided unprecedented information on both the intrinsic frequency spectrum and the probability distribution function of wave fluctuations.

The observed vertical displacements of the balloons, i.e. the recorded updraft and downdraft events, were randomly distributed and resulted to a major part from the interference of gravity waves. These up and downdrafts events showed a wide spectrum of amplitudes and updraft velocities. An important finding of Podglajen et al. (2016) is that updraft events with an amplitude of, e.g., 100 m occur by an order of magnitude more frequently than updraft events with an amplitude of around 200 m. The following conclusion can be drawn from the balloon observations. The most frequently occurring rather shallow updrafts do not produce any ice. Their amplitudes are too low. The ice saturation ratio $S_{i,on}$, necessary for ice nucleation, is not reached. The most frequently occurring updraft events, that contribute to ice production, are those with low amplitudes. However they lead to low amounts of ice crystals. The ice nucleation events caused by frequently occurring shallow updrafts thus dominate the cirrus characteristics including the ICNC levels. The strong updraft events with large amplitude, leading to large ICNC values of 300-1000 L$^{-1}$, are seldom, but occur from time to time. Another aspect of the strong decrease of the updraft occurrence frequency with increasing updraft amplitude is that the conditions for heterogeneous ice nucleation are much more favorable than for homogeneous freezing events, since the onset values of $S_{i,on}$ are lower in the case of heterogeneous ice nucleation, compared to homogeneous freezing onset values, so that lower amplitudes are sufficient to initiate heterogeneous ice nucleation. There are thus many opportunities (updraft events) to start heterogeneous ice nucleation, but only a few to initiate homogeneous freezing. As long as INPs are available in rising air parcels they control ice formation and heterogeneous ice nucleation dominates.

## 4   Summary

For the first time, an observational Arctic cirrus data set was presented that covers the entire winter half year from October to March. Lidar and cloud radar observations of aerosol and cloud profiles were performed aboard the German ice breaker *Polarstern* at latitudes >85°N as part of the MOSAiC expedition 2019-2020. The winter cirrus clouds were characterized in terms of geometrical, optical, and microphyscial properties. The ice clouds were optically thin with cloud mean extinction

coefficients mostly ranging from about 30-300 $Mm^{-1}$, and IWC and ICNC values frequently between 0.001 and 0.02 g $m^{-3}$
and 0.01-10 $L^{-1}$, respectively. In ice virga, typical ICNC values accumulated between 1-10 $L^{-1}$, in many cases values up
to 50 $L^{-1}$ occurred, however, ICNC values rarely exceeded 100 $L^{-1}$. The cirrus layers, observed from October 2019 to
March 2020, developed in a wildfire-smoke-polluted environment. We hypothesized that short-term updrafts with shallow
amplitudes were the reason for the observed low ICNC values.

Main goal of the data analysis in this part 1 was to provide observational evidence that the Siberian wildfire smoke (organic
aerosol particles) significantly influenced ice nucleation in the upper troposphere during the MOSAiC winter months. Three
observational findings support our hypothesis that the smoke pollution contributed or even dominated cirrus formation. These
arguments are summarized in Fig. 12. (1) We observed a highly polluted upper troposphere throughout the entire winter half
year. 101 clear sky PSAC observations from November 2019 to February 2020 are shown in Fig. 12. The observation of a
constant upper tropospheric aerosol load suggested that the INP reservoir was permanently refilled from above, i.e., from
lower stratosphere, so that a large amount of INPs was available to control ice nucleation processes in the upper troposphere
throughout the MOSAiC winter half year. The respective 532 nm particle extinction coefficients were 20 times higher than
the extinction coefficients for background aerosol conditions. (2) The observed smoke PSAC values of 10$\pm$5 $\mu m^2$ $cm^{-3}$
in the upper troposphere were high enough to dominate ice nucleation and widely suppress homogeneous freezing events.
The simulations in part 2 will provide more details. (3) The frequently observed maximum cirrus ice saturation ratios of
1.3-1.5, observed with radiosondes in extended cirrus fields at temperatures from $-60$ to $-75°$C, point to the dominance
of heterogeneous ice nucleation on inefficient INPs as it is expected when glassy smoke particles serve as INPs. This is
an important finding and corroborates the assumption that aged wildfire smoke alone was responsible for heterogeneous ice
nucleation. In the presence of efficient INPs, such as mineral dust particles, the maximum ice saturation ratios would probably
have been at all below 1.25 (Ullrich et al., 2017; Ansmann et al., 2019; Dekoutsidis et al., 2024). On the other hand, in cases
with dominating homogeneous freezing, the maximum ice saturation ratios should have been found mostly between 1.5 and
1.6 (Dekoutsidis et al., 2024).

Disregarding all these facts pointing to a strong impact of wildfire smoke on cirrus formation over the North Pole region
in the winter of 2019-2020, we cannot rule out that homogeneous freezing contributed to cirrus formation as well. It is a
reasonable option that a certain amount of air parcels were free or almost free of smoke INPs after numerous updraft events so
that conditions became favorable for homogeneous ice nucleation on background aerosol particles. In part 2, we will continue
our Arctic cirrus studies and present the key findings of the MOSAiC-related simulation studies.

## 5 Data availability

Polly lidar observations (level 0 data, measured signals) are in the PollyNet database (Polly, 2024). All the analysis products
are available at TROPOS upon request (polly@tropos.de) and at https://doi.pangaea.de/10.1594/PANGAEA.935539 (Ohneiser
et al., 2021). Cloud radar data are downloaded from the ARM data base (ARM, 2024; ARM-MOSAiC, 2024). MOSAiC ra-
diosonde data are available at https://doi.org/10.1594/PANGAEA.928656 (Maturilli et al., 2021, 2022) Backward trajectory

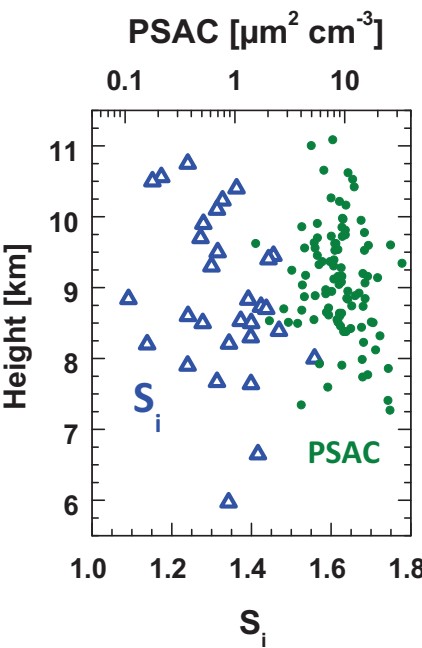

**Figure 12.** Maximum cirrus ice saturation ratios $S_{i,max}(z)$ (open blue triangles). $S_i(z)$ profiles measured with radiosondes, that ascended through 30 extended cirrus fields, were analyzed. In addition, particle surface area concentrations (PSAC, green circles) for the tropopause region, obtained from 101 clear sky lidar observations (November 2019 to February 2020) are shown.

analysis has been performed by air mass transport computation with the NOAA (National Oceanic and Atmospheric Administration) HYSPLIT (HYbrid Single-Particle Lagrangian Integrated Trajectory) model (HYSPLIT, 2024). LIRAS-ice products can be obtained on request (by contacting the corresponding author)

## 6 Author contributions

The paper was written and designed by AA and CJ. The aerosol and cloud data analysis was performed by CJ, JR, JB, KO, HG, JH, DA, TG, and PS. RE, HGr, MR, JH, and DA took care of the lidar observations aboard *Polarstern* during the one year MOSAiC expedition. SD was responsible for the data analysis of the *Polarstern* radiosonde observations and related quality assurance efforts. DAK and UW were involved in the interpretation of the findings. All coauthors were actively involved in the extended discussions and the elaboration of the final design of the manuscript.

## 7 Competing interests

Daniel A. Knopf is a member of the editorial board of Atmospheric Chemistry and Physics

## 8 Financial support

The Multidisciplinary drifting Observatory for the Study of the Arctic Climate (MOSAiC) program was funded by the German Federal Ministry for Education and Research (BMBF) through financing the Alfred Wegener Institut Helmholtz Zentrum für Polar und Meeresforschung (AWI) and the *Polarstern* expedition PS122 under grant N-2014-H-060_Dethloff. The lidar analysis on smoke-cirrus interaction was further supported by BMBF funding of the SCiAMO project (MOSAIC-FKZ 03F0915A). The radiosonde program was funded by AWI awards AFMOSAiC-1_00 and AWI_PS122_00, the U.S. Department of Energy Atmospheric Radiation Measurement Program, and the German Weather Service. This project has also received funding from the European Union's Horizon 2020 research and innovation program ACTRIS-2 Integrating Activities (H2020-INFRAIA-2014 - 2015, grant agreement no. 654109) as well as from the European Union's Horizon Europe Programme under Grant Agreement No. 101137639 (CleanCloud). We gratefully acknowledge the funding by the Deutsche Forschungsgemeinschaft (DFG, German Research Foundation) – project no. 268020496 - TRR 172, within the Transregional Collaborative Research Center "ArctiC Amplification: Climate Relevant Atmospheric and SurfaCe Processes, and Feedback Mechanisms (AC)3". DAK acknowledges support by U.S. Department of Energy's (DOE) Atmospheric System Research (ASR) program, Office of Biological and Environmental Research (OBER) (grant no. DE-SC0021034).

*Acknowledgements.* Data used in this article were produced as part of the international Multidisciplinary drifting Observatory for the Study of the Arctic Climate (MOSAiC) with the tag MOSAiC20192020 and the Project_ID: AWI_PS122_00. We would like to thank everyone who contributed to the measurements used here (Nixdorf et al., 2021). Radiosonde data were obtained through a partnership between the leading Alfred Wegener Institute, the Atmospheric Radiation Measurement user facility, a U.S. Department of Energy facility managed by the Biological and Environmental Research Program, and the German Weather Service (DWD). We would like to thank the RV *Polarstern* crew for their perfect logistical support during the one-year MOSAiC expedition.

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
