# Peer review of "Impact of wildfire smoke on Arctic cirrus formation, part 1: analysis of MOSAiC 2019-2020 observations"

_EGUsphere, 2024_

## Referee Comment (RC1)

**Review of: "Impact of wildfire smoke on Arctic cirrus formation, part 1: analysis of MOSAiC 2019-2020 observations" by Ansmann et al.**

**Content**

The study describes polar lidar obervations of cirrus clouds and aerosols in the time frame between October 2019 and March 2020 during the famos MOSAiC expedition. This manuscript is the first part (observations) of a series of two manuscripts. The lidar observations were combined with radar and in-situ particle observations. 20 cirrus events where analysed and the main findings are the low ice crytstal number concentrations (ICNC) point to heterogeneous ice nucleation, the evidence of elevated smoke pollution levels, and high ice saturation ratios point ot inefficient INPs.

**Overall impression and rating**

I'm reviewer of both manuscript parts of this paper series. The overall impression of this first part of the manuscripts is very good. In particular, this manuscript is written in a clear way and the important aspects are considered. It is well organized and the analysis and results are clearly structured and communicated. The study is very valuable for the scientific community because it shows rare observations of polar cirrus clouds which are potentially influenced by wildfire aerosols. For this reasons, I recommend publication in ACP after addressing my very few comments and a subsequent very minor manuscript revision.

**Main comments/questions:**

- Figure 7: ICNC obervations between 01:30 and 02:30 on the 22 January 2020 show high values around 50L-1 while below in the fall streak ICNC are one orders of magnitude lower. This is an indication for mulitple nucleation events and could be an indication of homogeneous nucleation as INPs are already consumed by previous nucleations. So it is hard to tell weather it is homogeneous or heterogeneous nucleation just from the microphysical observations wihtout having information of ICNC in the important nucleation layer above. It could be that most of the cirrus cases where formed via both nucleation pathways. First consuming all INPs and than subsequently homogeneous nucleation kicks in. In the virga you would just see the heterogeneously formed ice particles because of their larger sizes. So the ICNC values in the nucleation layer are more or less speculation as indicated also by your measurements line 358-365. I would recommend, in order to argue as evenly as possible, to mention this possible path in the discussion, because a final answer is not possible only from the observational side.

**Specific comments/questions:**

- Line 35-36: You list here just lidar observations in the references. However, there are also other observations of wildfire plumes available (e.g. obs. with OMPS by Kloss et al, 2017). I recommend to cite here a little bit more balanced.

- Line 44: "of the order of 500 L^-1 in the case of homogeneous freezing" - It is actually not true that homogeneous freezing is producing always high ICNC. For example under calm (w ~ 1-5cm/s) and warm conditions (e.g. 220K) you could have also much smaller ICNC with homogeneous freezing as can be seen in Fig. 5 of your cited paper Kärcher et al. 2022 or in another paper by Krämer et al. 2016 (Fig. 6). I recommand to just mention here a range of typical ICNC for freezing.

- Line 149: Maybe you can add here how the 1936 lidar profiles are distributed over the half year.

- Line 168: Can you give an estimation to which altitude maybe also realtive to the tropopause it was possible to apply the LIRAS-ice analysis scheme. This would help the reader to estimate how much of the cirrus was missed due to the limitations of the radar. If I understand correclty, you will miss always the nucleation zone directly at/below the tropopause for the microphysical property determination.

- Line 270: The difference might come from the different sites. Espacially at Kupio the influence of mountain waves genertated by the Scandinavian mountains might influence the ICNC and thus also the COT. This could be added as speculation here.

- Line 275-276: As I mentioned before, the simple discrimination between heterogeneous and homogeneous nucleated ice particles just by the number concentration is not always reliable. It is important to look at the cloud air mass history to determine the updraft speed during nucleation to better classify, if the ice partcles are formed heterogeneously and homogeneously (Krämer et al. 2016).

**Technical comments/suggestions:**

- Line 37: There is a Latex/Bibtex citation missing.

- Figure 2: This is a cool photo !

- Line 441: Reference of PollyNet database is missing in the PDF.

**References**

- Kärcher, B. (2022). A parameterization of cirrus cloud formation: Revisiting competing ice nucleation. Journal of Geophysical Research: Atmospheres, 127, e2022JD036907. https://doi.org/10.1029/2022JD036907

- Kloss, C., Berthet, G., Sellitto, P., Ploeger, F., Bucci, S., Khaykin, S., Jégou, F., Taha, G., Thomason, L. W., Barret, B., Le Flochmoen, E., von Hobe, M., Bossolasco, A., Bègue, N., and Legras, B.: Transport of the 2017 Canadian wildfire plume to the tropics via the Asian monsoon circulation, Atmos. Chem. Phys., 19, 13547–13567, https://doi.org/10.5194/acp-19-13547-2019, 2019.

- Krämer, M., Rolf, C., Luebke, A., Afchine, A., Spelten, N., Costa, A., Meyer, J., Zöger, M., Smith, J., Herman, R. L., Buchholz, B., Ebert, V., Baumgardner, D., Borrmann, S., Klingebiel, M., and Avallone, L.: A microphysics guide to cirrus clouds – Part 1: Cirrus types, Atmos. Chem. Phys., 16, 3463–3483, https://doi.org/10.5194/acp-16-3463-2016, 2016.

---

## Author Comment (AC1)

Dear reviewer,

thank you for reading the manuscript and for providing good and constructive suggestions. That helped a lot to improve the manuscript. We tried to consider most of the comments and suggestions.

All essential changes in the main text body are **marked in bold.**

First of all, a statement of the editor:

In the revised manuscript, I would like to ask you to take into account the following:
in the new ACP 'Guidelines for authors' ( https://www.atmospheric-chemistry-and-physics.net/policies/guidelines_for_authors.html ) it is recommended that the abstract should have fewer than 250 words - your abstract has far more words.

We shortened the abstract. Now we have 299 words. We followed the classical abstract structure: (1) goal and topic of the paper (2) methods/instruments/tools used, (3) key results.

**Reviewer 1**

Content

The study describes polar lidar obervations of cirrus clouds and aerosols in the time frame between October 2019 and March 2020 during the famos MOSAiC expedition.  This manuscript is the first part (observa- tions) of a series of two manuscripts. The lidar observations were combined with radar and in-situ particle observations. 20 cirrus events where analysed and the main findings are the low ice crytstal number con- centrations (ICNC) point to heterogeneous ice nucleation, the evidence of elevated smoke pollution  levels, and high ice saturation ratios point to inefficient INPs.

Overall impression and rating

I'm reviewer of both manuscript parts of this paper series. The overall impression of this first part of the manuscripts is very good. In particular, this manuscript is written in a clear way and the important aspects are considered.  It is well organized and the analysis and results are clearly structured and communicated. The study is very valuable for the scientific community because it shows rare observations of polar cirrus clouds which are potentially influenced by wildfire aerosols. For this reasons, I recommend publication in ACP after addressing my very few comments and a subsequent very minor manuscript revision.

Main comments/questions:

• Figure 7: ICNC observations between 01:30 and 02:30 on the 22 January 2020 show high values around 50L-1 while below in the fall streak ICNC are one orders of magnitude lower. This is an indication for multiple  nucleation events and could be an indication  of homogeneous nucleation  as INPs are already consumed by previous nucleations. So it is hard to tell weather it is homogeneous or heterogeneous nucleation just from the microphysical  observations without having information of ICNC in the important nucleation layer above.  It could be that most of the cirrus  cases where formed via both nucleation pathways. First consuming all INPs and than subsequently homogeneous nucleation kicks in. In the virga you would just see the heterogeneously formed ice particles because of their larger sizes. So the ICNC values in the nucleation  layer are more or less speculation  as indicated also by your measurements line 358-365. I would recommend, in order to argue as evenly as possible,  to mention this possible path in the discussion,  because a final answer is not possible only from the observational side.

These statements are hypotheses and are not supported by our MOSAiC observations. Our observations tell us another story. Before we start with the step-by-step reply to all questions and comments, it makes sense to begin with the following two cirrus scenarios so that our position becomes very clear.

Two different scenarios regarding cirrus formation in the upper troposphere can be distinguished:

Scenario 1: Cirrus evolution takes place in an isolated air mass that may contain a lot of INPs in the beginning of the cirrus evolution (that may have started hours or even days before the cirrus containing air mass reached and crossed, e.g., Polarstern). Ice nucleation takes place in the uppermost part of the cirrus layer. During each air-parcel lofting event, ice nucleation occurs and the number of INPs in the INP reservoir continuously decreases in the given air parcel. After, e.g., 100 air parcel lofting events, the INP reservoir may be empty so that during further updraft events, homogeneous freezing of sulfate particles is the only ice nucleation process left.

This scenario 1 is described in almost all cirrus simulation papers. But this scenario 1 does often not reflect the real world! We observed in many cases, e.g., when cirrus developed in Saharan dust layers (Ansmann et al., ACP, 2019) that the INP reservoir was never empty.

Therefore we need to introduce a second scenario. This scenario 2 is in agreement with our numerous measurements (with cirrus in dust or in smoke layers). Regarding smoke, we point to cirrus formation in Californian smoke over Cyprus (Mamouri et al., 2023) and in Australian smoke (over Punta Arenas in the whole year of 2020), and now also to cirrus formation in Siberian smoke (MOSAiC). The following scenario was always found:

Scenario 2: Cirrus development takes place in an upper tropospheric layer filled with smoke INPs. The main smoke reservoir is above the tropopause and the upper tropospheric layer is refilled from the stratospheric smoke reservoir. There is a continuous flux of particles from the lower stratosphere into the upper troposphere and then a flux further down so that the pollution level in the upper troposphere in constant (as observed during MOSAiC over the entire winter half year). At such conditions, the upper tropospheric INP reservoir is always well filled and ice nucleation during updraft events (close to cirrus top) are not able to significantly reduce the INP concentration (in the reservoir). Losses of INPs during ice nucleation events in a given air parcel are compensated by refilling from above. Even after 100 air-parcel lofting events the situation concerning available INPs remains almost unchanged.

As mentioned before, this scenario agrees well with the MOSAiC lidar observations of aerosol and cirrus layers during the entire winter half year. In Sect. 3 in part 2, we present a new figure (Fig 4) that corroborates that there was always smoke in the top region of cirrus. The best example (most easy to see) for the continuous presence of smoke in the ice nucleation zones (in the upper part of the cirrus) was shown in Fig. 14 in Ansmann et al. (2023) covering the MOSAiC cirrus period from 25-29 February 2020.

We emphasize that our findings support the hypotheses that heterogeneous ice nucleation on smoke particles had a clear impact on ice nucleation. However, we agree and thus follow the recommendation of the reviewer and state that there is always the option that homogeneous freezing sets in when the INP reservoir is empty.

Nevertheless, we also ask the question in the manuscript: How reasonable is the assumption that favorable conditions for homogeneous freezing (absolute clean conditions, only the background aerosol is left) were given during the MOSAiC winter months in view of the constantly high levels of

smoke pollution and the always available option of refilling the INP reservoir from above (from the stratosphere) over all six winter months?

Motivated by the comment that 'ICNC values in the nucleation zone estimated from ICNC observation in virga zone about 3-4 km below the nucleation zone are simply speculation', we added a discussion in Sect. 3.4 on the impact of ice-ice-collision and aggregation processes on the ICNC height profile. The study of Kienast-Sj\"ogren, Spichtinger and Gierens (ACP, 2013) shows that the aggregation impact on ICNC is negligible at low Artic temperatures of -50, -60, -70°C. We should accept that! Wolf et al. (2018) corroborate this. They show balloon observations of a height-independent ICNC profile from the lower part of virga at 6 km up to cirrus top at 11.5 km in the case of an Arctic cirrus measured in February 2016. ICNC values were between 1 and 10 L-1 and showed practically no height dependence. So, these observations are in line with the study of Kienast-Sj\"ogren et al. (2013). In contrast, Field and Heymsfield (2003) and Mitchell et al. (2018) point to a potentially large impact of aggregation processes on ICNC, but for significantly higher ICNCs. So, to our opinion, a reasonable assumption in case of our MOSAiC observation is that we should not ignore aggregation-related ICNC decreases with increasing distance from cirrus top, but in the majority of cases the effect is quite small, so that the ICNC values in virga can be used as estimates of ICNC values in the cirrus top region.

Specific comments/questions:

• Line 35-36: You list here just lidar observations in the references. However, there are also other observations of wildfire plumes available (e.g. obs. with OMPS by Kloss et al, 2017). I recommend to cite here a little bit more balanced.

We include Kloss et al. (2017).

• Line 44: "of the order of 500 L^-1 in the case of homogeneous freezing" - It is actually not true that homogeneous freezing is producing always high ICNC. For example under calm (w ~ 1-5cm/s) and warm conditions (e.g. 220K) you could have also much smaller ICNC with homogeneous freezing as can be seen in Fig. 5 of your cited paper Kärcher et al. 2022 or in another paper by Krämer et al. 2016 (Fig. 6). I recommand to just mention here a range of typical ICNC for freezing.

We removed such comments at all, as mentioned above. We agree that ICNC numbers do not indicate the nucleation mode. The simulations (in part 2) corroborate that after adding a simulation with large-scale lofting showing that even homogeneous freezing events can cause low ICNCs. Both modes can produce low and high ICNC numbers. We state that in the conclusions of part 2.

• Line 149: Maybe you can add here how the 1936 lidar profiles are distributed over the half year.

We added these numbers in Sect. 2.3.

• Line 168: Can you give an estimation to which altitude maybe also relative to the tropopause it was possible to apply the LIRAS-ice analysis scheme. This would help the reader to estimate how much of the cirrus was missed due to the limitations of the radar. If I understand correclty, you will miss always the nucleation zone directly at/below the tropopause for the microphysical property determination.

We do not like this idea! On 6 Dec 2019 (in Fig.9b), the radar was able to cover almost the entire cirrus height range (up to 8.5 km). This holds for many cases in November and December when cirrus top heights were often at 8-9 km. In other cases with very low ICNC < 10 L-1, aggregation effects are probably of minor impact so that the distance between 8 km and cirrus top at 10-11 km height has no big impact on the estimation of ICNC in the cirrus top range. The study of Kienast-

Sj\"ogren, Spichtinger and Gierens (ACP, 2013) support this assumption, and the balloon observation of Wolf et al. (2018) are in line with this. As mentioned, we discuss the aggregation impact in detail in Sect. 3.4. The Sect. 3.4 is new! The introduction of this specific section was necessary to better highlight the importance of ICNC observations in the virga. These virga ICNC values can be regarded as the link between part 1 (observations) and part 2 (simulations).

In Sect. 3.3, we write: We assume that each well-resolved virga is linked to a singular, individual ice nucleation event so that the virga occurrence frequency is equal or almost equal to the occurrence frequency of updraft and associated ice nucleation events. The retrieved virga ICNC and the observed temporal width and structural complexity of the virga contain information about updraft strength, duration, and complexity (individual or cluster-like occurrence) and the associated ice nucleation intensity. The short-term updraft periods during which the nucleation events develop occur randomly and are omnipresent in the upper troposphere as a result of gravity wave activity (Podglajen et al., 2016}, windshear-induced turbulence production, and orographic influences.

In this context, we also added a more general discussion of the balloon observations of Podglajen et al. (GRL, 2016) in Sect. 3.5. This addition was motivated by the question: Why were the observed ICNC values typically so low? (see page 21 in Sect. 3.5). Podglajen et al. (2016) clearly showed that shallow updrafts with relatively small amplitudes (producing low amounts of ice crystals according to the simulations in part 2) occur much more frequently in the UTLS than strong updrafts with relatively large amplitudes (able to produce large amounts of crystals). So, the dominance of shallow updrafts is the reason for the usually found low ICNCs, we concluded. According to Podglajen et al. (2016) updrafts with amplitudes, e.g., of 100 m occur an order magnitude more frequently than updrafts with an amplitude of 200 m. According to this, there are many opportunities (i.e. updraft events) to initiate ice nucleation and to produce low amounts of ice crystals, but only occasionally stronger updraft events that permit the nucleation of a large number of ice crystals and lead to large ICNCs.

• Line 270: The difference might come from the different sites. Especially at Kupio the influence of mountain waves generated by the Scandinavian mountains might influence the ICNC and thus also the COT. This could be added as speculation here.

Thank you for this hind. We added this information.

• Line 275-276: As I mentioned before, the simple discrimination between heterogeneously and homogeneously nucleated ice particles just by the number concentration is not always reliable. It is important to look at the cloud air mass history to determine the updraft speed during nucleation to better classify, if the ice particles are formed heterogeneously and homogeneously (Krämer et al., 2016).

As mentioned we removed statements on the link between nucleation mode and expected ICNC.

Regarding the use of air mass history analyses: As mentioned in the reply letter to the review of part 2, we think that short-term updraft events are mainly responsible for ice nucleation. These short-term updrafts occur randomly and are not resolved by modelling. Air mass history studies make sense in cases where large-scale lofting dominates and initiates ice formation.

Regarding the statement '…if the ice particles are formed heterogeneously and homogeneously', we are always puzzled …. and ask ourself: How can long-lasting large scale lofting in smoke polluted air lead, at the end, to favorable conditions for homogeneous ice nucleation where an absolutely INP-free air mass is required (only background aerosol is left) before homogeneous freezing can take place?

Technical comments/suggestions:

• Line 37: There is a Latex/Bibtex citation missing.

Is improved.

• Figure 2: This is a cool photo !

YES!

• Line 441: Reference of PollyNet database is missing in the PDF.

We improved that.

---

## Referee Report (RR1)

**Review of: "Impact of wildfire smoke on Arctic cirrus formation, part 1: analysis of MOSAiC 2019-2020 observations" by Ansmann et al.**

Thank you very much for revising the manuscript and addressing my comments. I think that the manuscript has improved significantly as a result of the revision. In particular, I find the discussion even more balanced and the hypotheses better explained.

- I support the general statement that heterogeneous freezing played a major role during the period. The only thing I still don't quite agree with is the idea that homogeneous nucleation should only occur when the INP reservoir is completely empty. Homogeneous nucleation can also occur in parallel with heterogeneous nucleation if the updrafts are strong enough and the supersaturation is getting large enough (see, for example, Krämer et al. 2016 or Rolf et al. 2012 or Kärcher et al. 2022). This happens in the case when the existing ice surface after/during heterogeneous freezing is not sufficient to reduce the ice supersaturation via diffusive growth of the ice crystals. The short term updrafts by gravity waves of up to ~1 m/s reported in Podglajen et al. are sufficient to add homogeneous freezing even in the presence of INP particles (see for instance Fig.7 in Rolf et al. 2012 or Kärcher et al Sec. 4.3). In Kärcher et al., only warm conditions were investigated. For cold conditions, as present in this study, I would expect the supersaturation to respond even more strongly to an additional updraft and even more likely to initiate homogeneous freezing because the saturated vapor pressure has a strong temperature dependence. The statement of "homogeneous nucleation should only occur when the INP reservoir is completely empty" is not particularly relevant to the first part of the paper series. I leave it to the editor to decide whether this needs to be explained further in Part 1 of the manuscript or it should be better mentioned in Part 2.

- Another small thing is, that you should avoid the very strong wording of "unlimited INPs" see line 496. Please change it to something like "large" or "virtually unlimited"... Otherwise, it would exist forever. I also believe that the INP reservoir is large, but it is definitely not infinite.

- Otherwise, I have no further objections to this part regarding publication in ACP.

**Technical comments:**

- Line 456: Please remove "the" in "a the updraft event"

---

## Author Response (AR2)

Dear Editor, dear reviewer!

thank you again for careful reading and the comments. We considered them in the revised version.

Ref#1:

1) I support the general statement that heterogeneous freezing played a major role during the period. The only thing I still don't quite agree with is the idea that homogeneous nucleation should only occur when the INP reservoir is completely empty. Homogeneous nucleation can also occur in parallel with heterogeneous nucleation if the updrafts are strong enough and the supersaturation is getting large enough (see, for example, Krämer et al. 2016 or Rolf et al. 2012 or Kärcher et al. 2022). This happens in the case when the existing ice surface after/during heterogeneous freezing is not sufficient to reduce the ice supersaturation via diffusive growth of the ice crystals. The short term updrafts by gravity waves of up to ~1 m/s reported in Podglajen et al. are sufficient to add homogeneous freezing even in the presence of INP particles (see for instance Fig.7 in Rolf et al. 2012 or Kärcher et al Sec. 4.3). In Kärcher et al., only warm conditions were investigated. For cold conditions, as present in this study, I would expect the supersaturation to respond even more strongly to an additional updraft and even more likely to initiate homogeneous freezing because the saturated vapor pressure has a strong temperature dependence. The statement of "homogeneous nucleation should only occur when the INP reservoir is completely empty" is not particularly relevant to the first part of the paper series. I leave it to the editor to decide whether this needs to be explained further in Part 1 of the manuscript or it should be better mentioned in Part 2.

We agree and changed the text accordingly.

On  page 15 (given in red), we write:

…. will not lead to an empty or almost depleted INP reservoir so that favorable conditions for homogeneous freezing were given ….

On page 16 we write:

However, even in the case of a highly polluted UTLS we cannot completely rule out that occasionally situations occurred in which almost smoke-free conditions were given in a number of air parcels after the consumption of all or most of the smoke INPs so that homogeneous freezing became an additional ice nucleation option. As shown by Rolf et al. (2012), Krämer et al. (2016) and Kärcher et al. (2022), homogeneous ice nucleation is possible in the presence of a low number of INPs when the updrafts are strong enough so that the diffusional growth of the few heterogeneously nucleated ice crystals by water vapor deposition is not sufficient to reduce the relative humidity in the ascending air parcel significantly and to prevent that the ice saturation ratio $S_i$ in the lofted air parcel reaches and exceeds the onset ice saturation ratio $S_{i,on}$ for homogeneous freezing.

On page 22 (summary) we write:

It is a reasonable option that a certain amount of air parcels were free or almost free of smoke INPs after numerous updraft events so that conditions became favorable for homogeneous ice nucleation on background aerosol particles.

2) Another small thing is, that you should avoid the very strong wording of "unlimited INPs" see line 496. Please change it to something like "large" or "virtually unlimited"... Otherwise, it would exist forever. I also believe that the INP reservoir is large, but it is definitely not infinite.

We change that on page 22.

3) Line 456: Please remove "the" in "a the updraft event"

Done!